# Enzyme activity and selectivity filter stability of ancient TRPM2 channels were simultaneously lost in early vertebrates

Iordan Iordanov[1,2†], Balázs Tóth[1,2†], Andras Szollosi[1,2], László Csanády[1,2*]

[1]Department of Medical Biochemistry, Semmelweis University, Budapest, Hungary; [2]MTA-SE Lendület Ion Channel Research Group, Semmelweis University, Budapest, Hungary

**Abstract** Transient Receptor Potential Melastatin 2 (TRPM2) is a cation channel important for the immune response, insulin secretion, and body temperature regulation. It is activated by cytosolic ADP ribose (ADPR) and contains a nudix-type motif 9 (NUDT9)-homology (NUDT9-H) domain homologous to ADPR phosphohydrolases (ADPRases). Human TRPM2 (hsTRPM2) is catalytically inactive due to mutations in the conserved Nudix box sequence. Here, we show that TRPM2 Nudix motifs are canonical in all invertebrates but vestigial in vertebrates. Correspondingly, TRPM2 of the cnidarian *Nematostella vectensis* (nvTRPM2) and the choanoflagellate *Salpingoeca rosetta* (srTRPM2) are active ADPRases. Disruption of ADPRase activity fails to affect nvTRPM2 channel currents, reporting a catalytic cycle uncoupled from gating. Furthermore, pore sequence substitutions responsible for inactivation of hsTRPM2 also appeared in vertebrates. Correspondingly, zebrafish (*Danio rerio*) TRPM2 (drTRPM2) and hsTRPM2 channels inactivate, but srTRPM2 and nvTRPM2 currents are stable. Thus, catalysis and pore stability were lost simultaneously in vertebrate TRPM2 channels.
DOI: https://doi.org/10.7554/eLife.44556.001

**\*For correspondence:**
csanady.laszlo@med.semmelweis-univ.hu

†These authors contributed equally to this work

## Introduction

The Transient Receptor Potential Melastatin 2 (TRPM2) protein forms $Ca^{2+}$-permeable non-selective cation channels that are expressed in immune cells, pancreatic beta cells, cardiomyocytes, and neurons in the brain (*Nagamine et al., 1998*; *Perraud et al., 2001*; *Togashi et al., 2006*). TRPM2 channels become activated under conditions of oxidative stress (*Hara et al., 2002*) and contribute to the $Ca^{2+}$ influx that triggers insulin secretion (*Uchida et al., 2011*), immune cell activation (*Yamamoto et al., 2008*), and heat responses of heat sensitive neurons in the preoptic area of the hypothalamus responsible for body temperature regulation (*Song et al., 2016*). On the other hand, $Ca^{2+}$ influx through TRPM2 channels contributes to neuronal cell death following brain ischemia and in certain neurodegenerative diseases (*Hara et al., 2002*; *Kaneko et al., 2006*; *Fonfria et al., 2005*; *Hermosura et al., 2008*; *McQuillin et al., 2006*). Thus, TRPM2 has become an attractive pharmacological target for treating chronic inflammatory diseases, diabetes, congenital hyperinsulinism, excessive fever, and neuronal cell death following stroke.

Each subunit of a homotetrameric TRPM2 channel is formed by ~1500 amino acid residues that are organized into a large cytosolic N-terminal region, a pore forming transmembrane region with a topology typical to voltage-gated cation channels, and a cytosolic C-terminal region. The N-terminal region falls into three subdomains. In the C-terminal region, the conserved TRP helices and coiled-coil region are followed by an ~270 amino acid domain termed NUDT9-H, for homology with the soluble mitochondrial enzyme NUDT9 (*Perraud et al., 2001*), an ADP-ribose (ADPR) hydrolase (ADPRase) which splits ADPR into AMP and ribose-5-phosphate (*Perraud et al., 2003*). TRPM2

channels are co-activated by binding of cytosolic ADPR and $Ca^{2+}$ (*McHugh et al., 2003*; *Csanády and Törocsik, 2009*), but their activity also requires the presence of phosphatydil-inositol-bisphosphate ($PIP_2$) in the membrane (*Tóth and Csanády, 2012*).

The NUDT9-H domain of human TRPM2 binds ADPR (*Grubisha et al., 2006*) but is enzymatically inactive (*Iordanov et al., 2016*). ADPRase enzymes belong to the large family of Nudix (nucleoside diphosphate linked moiety X) hydrolases that harbor a highly conserved 'Nudix box' sequence (consensus: REUXEE, U = hydrophobic) at the heart of a more extended 'Nudix motif' (*Mildvan et al., 2005*). In the structure of the mitochondrial NUDT9 protein (Nudix box: REFGEE), the side chains of the first and third conserved glutamate (underlined) are both seen to participate in the formation of salt bridges that stabilize the active site, and the first glutamate is also involved in the coordination of a catalytic $Mg^{2+}$ ion (*Shen et al., 2003*); mutant NUDT9 proteins with RILGEE or REFGKK Nudix box sequences are inactive (*Perraud et al., 2003*; *Shen et al., 2003*), just as the NUDT9-H domain of human TRPM2 (*Iordanov et al., 2016*) which lacks the $Mg^{2+}$ coordinating glutamate (Nudix box: RILRQE).

Recent electron-cryomicroscopy structures of sea anemone (*Zhang et al., 2018*), zebrafish (*Huang et al., 2018*), and human (*Wang et al., 2018*) TRPM2 revealed an overall organization similar to that of other TRPM family channels in which the N-terminal regions, together with the C-terminal TRP and coiled-coil helices, assemble into a two-layered cytosolic structure below the transmembrane domain (e.g. *Autzen et al., 2018*; *Guo et al., 2017*; *Winkler et al., 2017*; *Yin et al., 2018*). In the zebrafish and human TRPM2 structures, an additional cytosolic layer formed by the four NUDT9-H domains is clearly resolved (*Huang et al., 2018*; *Wang et al., 2018*). Interestingly, TRPM2 channels from the sea anemone *Nematostella vectensis* (nvTRPM2) remain activatable by ADPR following deletion of the NUDT9-H domain, suggesting the presence of an additional ADPR binding site (*Kühn et al., 2016*). Correspondingly, in the ADPR-bound structure of zebrafish TRPM2, a clear density for a bound ADPR molecule is seen in a cleft formed by the first N-terminal domain (the 'N-terminal site'). The relative contributions of the N- and C-terminal ADPR binding sites to channel activation are still controversial. Although due to limited resolution the presence of ADPR bound to the NUDT9-H domain could be confirmed neither in the zebrafish nor in the human structure, in both structures pore opening was seen to be coupled to large conformational rearrangements of the four NUDT9-H domains. Moreover, channel activity was diminished by mutations in the N-terminal site in zebrafish, but not human, TRPM2, whereas it was abolished for both proteins by deletion of the NUDT9-H domain. Thus, the precise roles of the two types of ADPR binding site in TRPM2 channel activation remain unclear (*Huang et al., 2018*; *Wang et al., 2018*).

Interestingly, the Nudix box sequences are canonical for all invertebrate TRPM2 proteins, but their enzymatic activities have never been directly tested. In intact cells, deletion or mutations of the NUDT9-H domain sensitized nvTRPM2 currents toward activation by $H_2O_2$, which was interpreted to reflect ADPRase activity of the intact protein (*Kühn et al., 2016*). Here, we expressed and purified both full-length nvTRPM2 protein and its isolated NUDT9-H domain (nvNUDT9-H; Nudix box: AEFGEE), as well as full-length TRPM2 protein from the choanoflagellate *Salpingoeca rosetta* (srTRPM2) and its isolated NUDT9-H domain (srNUDT9-H; Nudix box: REFMEE), and found robust ADPRase activity for all four proteins. We then investigated how manipulations that abolish nvNUDT9-H enzymatic activity affect channel function. We further noticed that mutations in the selectivity filter that are responsible for the inactivation of human TRPM2 (*Tóth and Csanády, 2012*) are also absent in invertebrates, and therefore compared inactivation properties for two invertebrate and two vertebrate TRPM2 channel orthologs.

## Results

### nvTRPM2 is a true channel-enzyme, and isolated nvNUDT9-H recapitulates its catalytic properties

Sequence alignment of NUDT9-H domains of TRPM2 channel orthologs from unicellular flagellates to mammals (*Figure 1*, *right*) reveals that the deleterious EF→IL substitution in the Nudix box appeared between chordates and vertebrates: the Nudix box sequences of all invertebrates are canonical (*Figure 1*, *right*, *blue sequences*), whereas those of all vertebrates are vestigial (*Figure 1*, *right*, *red sequences*). This observation suggested to us that invertebrate TRPM2 channels might be

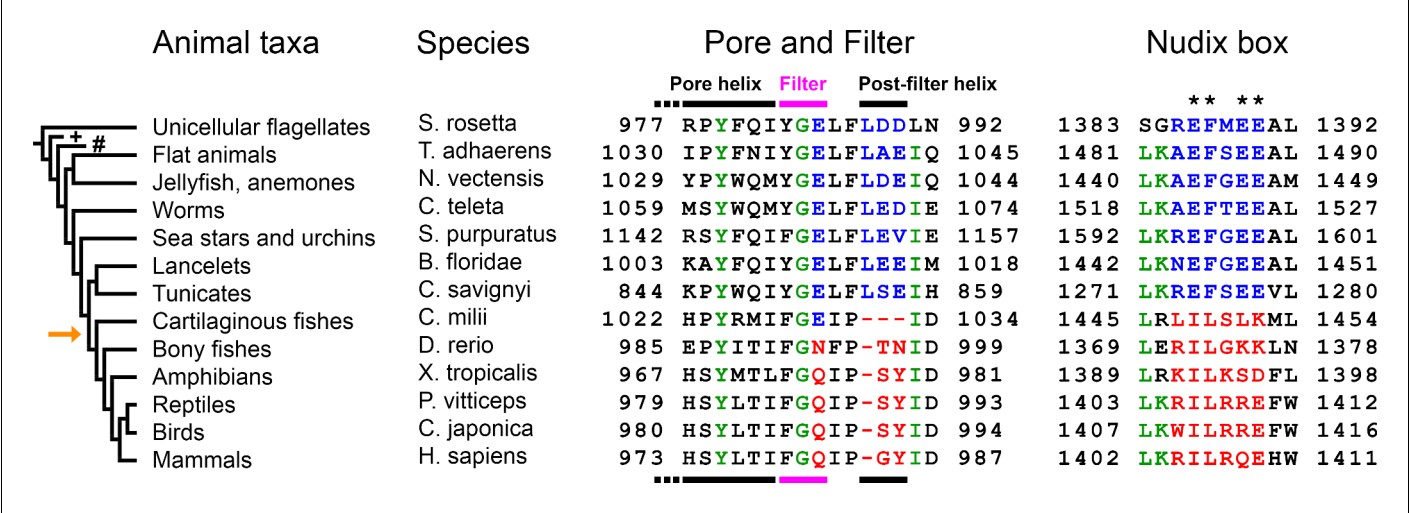

**Figure 1.** Evolution of the TRPM2 proteins in the *Animalia* kingdom. Evolutional progression through major animal taxa is indicated to the left, *orange arrow* marks the appearance of vertebrae. Chosen TRPM2 sequences from the indicated species are aligned in their Pore helix/Filter/Post-filter helix region (*left*, respective structural segments identified by bars on top and bottom) and Nudix box region (*right*); respective residue numbering is shown. The taxa *Porifera* (sponges; marked with '+') and *Ctenophora* (comb jellies; marked with '#') did not return any TRPM2-like sequences in BLAST. Asterisks mark Nudix box residues critical for ADPR hydrolysis. Concerted changes that happened between chordates and vertebrates in the sequences of the filter, post-filter helix, and the Nudix box are highlighted in *blue* (invertebrates) and *red* (vertebrates). Several of the listed proteins are predicted or uncharacterized proteins. For additional details on the chosen sequences, see Table in *Figure 1—figure supplement 1*.

DOI: https://doi.org/10.7554/eLife.44556.002

The following source data and figure supplement are available for figure 1:

**Source data 1.** Evolution of the TRPM2 proteins in the *Animalia* kingdom.
DOI: https://doi.org/10.7554/eLife.44556.004

**Figure supplement 1.** Evolutional progression of the TRPM2 proteins through the *Animalia* kingdom and the associated changes in their Pore/Filter and Nudix box regions.
DOI: https://doi.org/10.7554/eLife.44556.003

active chanzymes. The TRPM2 protein of the sea anemone *Nematostella vectensis* (nvTRPM2) has been shown to form functional channels activated by ADPR (*Kühn et al., 2015*) and is readily expressed and purified in large quantities (*Zhang et al., 2018*). To address its catalytic activity, we expressed full-length nvTRPM2 in HEK-293S cells and purified the detergent-solubilized protein to homogeneity (Materials and methods; *Figure 2—figure supplement 1A*). The protein was tested for ADPRase activity using a sensitive coupled enzymatic assay (Materials and methods), based on the colorimetric detection of inorganic phosphate ($P_i$) released from both ADPRase products (AMP and ribose-5-phosphate) by co-applied alkaline phosphatase. Because Nudix-family enzymes require the presence of $Mg^{2+}$ ions and basic pH for maximal activity (*Perraud et al., 2003*; *Mildvan et al., 2005*; *Iordanov et al., 2016*), the assay was done at pH = 8.0, in the presence of 10 mM $Mg^{2+}$. The purified nvTRPM2 protein showed robust ADPRase activity, characterized by a $K_M$ of 18 ± 2 µM (*Figure 2A*, *black symbols* and *fit line*) and a $k_{cat}$ of 41 ± 2 $s^{-1}$/subunit (*Figure 2B*, *black bar*), establishing nvTRPM2 as a true chanzyme.

To obtain a soluble model system of the nvTRPM2 enzymatic domain, we expressed isolated nvNUDT9-H in *E. coli*, and purified the domain to homogeneity (Materials and methods; *Figure 2—figure supplement 1B*). When assayed under similar conditions as the full-length protein, nvNUDT9-H displayed similarly robust ADPRase activity, with a somewhat lower $K_M$ (7.7 ± 1.4 µM; *Figure 2A*, *green symbols* and *fit line*) but a nearly identical $k_{cat}$ value (40 ± 3 $s^{-1}$; *Figure 2B*, *green bar*). Thus, although an additional binding site for ADPR is likely formed by the N-terminal domains also in nvTRPM2 (*Kühn et al., 2016*; *Huang et al., 2018*), hydrolysis of ADPR is mediated by the NUDT9-H domain which, when expressed in isolation, provides a convenient model system to study nvTRPM2 catalytic properties.

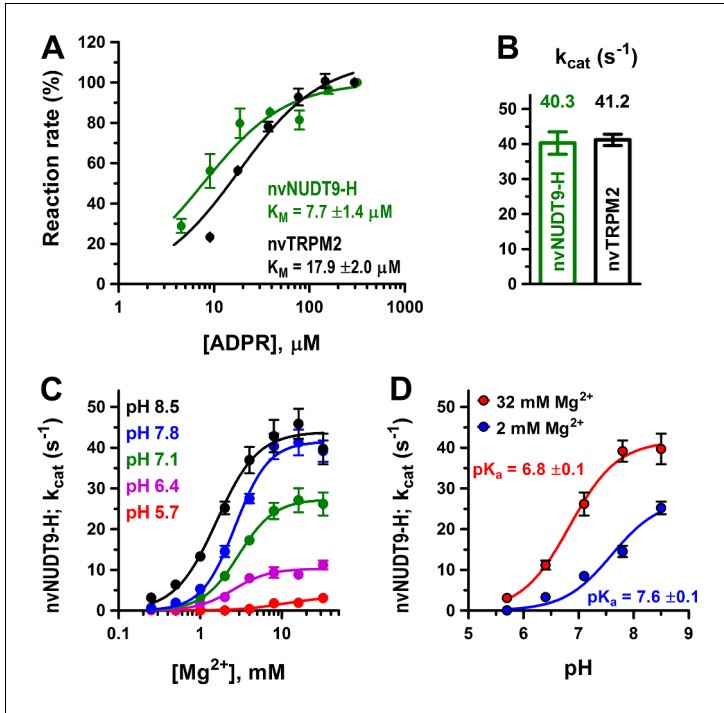

**Figure 2.** Enzymatic activity of full-length nvTRPM2 and of the isolated nvNUDT9-H domain. (A) Rates of ADPR hydrolysis by nvTRPM2 (*black*) and nvNUDT9-H (*green*) as a function of [ADPR], normalized to the rate measured at the highest ADPR concentration (free [$Mg^{2+}$] 10 mM, pH 8.0; see Materials and methods for details). *Solid lines* are fits to the Michaelis-Menten equation, $K_M$ values are indicated. (B) Estimated $k_{cat}$ values ($s^{-1}$) for nvTRPM2 (*black*, calculated per subunit of protein) and nvNUDT9-H (*green*), determined in the presence of saturating ADPR. Data are shown as mean ± SEM of at least 5 experiments. (C) $Mg^{2+}$-dependence of $k_{cat}$ ($s^{-1}$) for nvNUDT9-H, determined at several fixed pH values (*colors*) in the presence of saturating ADPR. *Solid lines* are fits to the Hill equation, yielding $K_{0.5}$ values of 10.8 ± 3.5 mM (pH 5.7, *red*), 2.5 ± 0.4 mM (pH 6.4, *magenta*), 3.0 ± 0.3 mM (pH 7.1, *green*), 2.7 ± 0.2 mM (pH 7.8, *blue*) and 1.6 ± 0.2 mM (pH 8.5, *black*); Hill coefficient was ~2 in each case. (D) pH-dependence of $k_{cat}$ ($s^{-1}$) for nvNUDT9-H, determined at two different fixed free [$Mg^{2+}$] (*colors*). *Solid lines* are fits to the equation $k_{cat} = k_{cat;max}/(1 + 10^{(pKa-pH)})$, with the calculated $pK_a$ values indicated. Data are shown as mean ± SEM of at least 3 experiments. See also *Figure 2—figure supplement 1*.

DOI: https://doi.org/10.7554/eLife.44556.005

The following source data and figure supplement are available for figure 2:

**Source data 1.** Enzymatic activity of full-length nvTRPM2 and of the isolated nvNUDT9-H domain.
DOI: https://doi.org/10.7554/eLife.44556.007
**Figure supplement 1.** Purification of nvTRPM2 and nvNUDT9-H.
DOI: https://doi.org/10.7554/eLife.44556.006

## $Mg^{2+}$- and pH-dependence of nvNUDT9-H catalytic activity

To obtain further insight into the catalytic mechanism of nvNUDT9-H, we systematically assessed the dependence of its molecular turnover rate on free [$Mg^{2+}$] and pH. At all pH values, $k_{cat}$ was a saturable function of free [$Mg^{2+}$] (*Figure 2C*, *colored symbols*), but fits to the Hill equation (*Figure 2C*, *colored curves*) reported a progressive reduction in maximal $k_{cat}$ ($k_{cat,max}$), and a progressive increase in $K_{1/2}$, at lower pH values (from $k_{cat,max} = 43.9 ± 1.9 \ s^{-1}$ and $K_{1/2} = 1.6 ± 0.2$ mM at pH 8.5 (*black curve*) to $k_{cat,max} = 3.5 ± 0.6 \ s^{-1}$ and $K_{1/2} = 10.8 ± 3.5$ mM at pH 5.7 (*red curve*)). On the other hand, the apparent Hill coefficient remained ~2 regardless of pH, suggesting the involvement of at least two $Mg^{2+}$ ions in ADPR coordination in the active site, as observed in structures of other ADPRases (*Gabelli et al., 2002*; *Shen et al., 2003*). In contrast, at any fixed [$Mg^{2+}$], pH-dependence of $k_{cat}$ (*Figure 2D*, *symbols*) was well described by titration of a single group (*Figure 2D*, *curves*), suggesting a key role in catalysis of a protonatable side chain. Furthermore, based on the sensitivity of its

apparent $pK_a$ value to free $[Mg^{2+}]$ (*Figure 2D*, *red* vs. *blue curve*), that side chain is likely near the bound $Mg^{2+}$ ion(s).

## srTRPM2 is also an active enzyme

To verify that enzymatic activity is not a unique property of nvTRPM2, we also expressed and purified the isolated NUDT9-H domain of the choanoflagellate *Salpingoeca rosetta* (srNUDT9-H) (Materials and methods, *Figure 3—figure supplement 1B*). Indeed, the purified srNUDT9-H protein also displayed clear ADPRase activity, with a $K_M$ value (5.9 ± 0.7 µM; *Figure 3A*, *black symbols* and *fit line*) similar to that of nvNUDT9-H, but a $k_{cat}$ value of only ~2 s$^{-1}$ in 1–10 mM $Mg^{2+}$. We could also clearly demonstrate $Mg^{2+}$ and pH dependence of $k_{cat}$, but the impact of these parameters on catalytic activity was more complex for srNUDT9-H compared to nvNUDT9-H (*Figure 3B*). At both pH 8.5 and 7.1, fitting the $[Mg^{2+}]$ dependence of $k_{cat}$ (*Figure 3B*, *black and green symbols*) required postulation of high- and low-affinity $Mg^{2+}$ binding sites. Fits to a double Hill-curve (*Figure 3B*, *solid curves*) returned an apparent Hill coefficient of ~2 for the high-affinity site and ~1 for the low-affinity site, suggesting high-affinity binding of two $Mg^{2+}$ ions to be required for catalysis, but further enhancement of $k_{cat}$ through low-affinity binding of a third $Mg^{2+}$. Similarly to nvTRPM2, $K_{1/2}$ of the high-affinity $Mg^{2+}$ site in srNUDT9-H was sensitive to pH ($K_{1/2}$ was 0.031 ± 0.002 mM at pH = 8.5 but 0.102 ± 0.008 mM at pH 7.1), consistent with the presence of a protonatable side chain near the

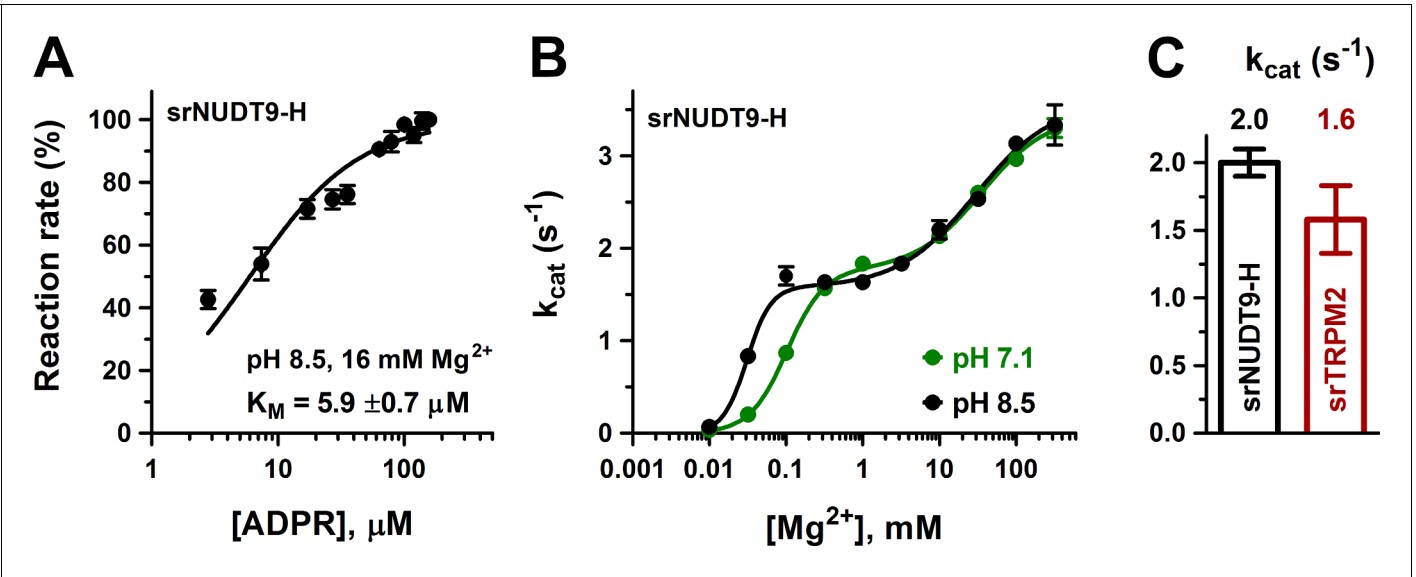

**Figure 3.** Enzymatic activity of full-length srTRPM2 and of the isolated srNUDT9-H domain. (**A**) Rate of ADPR hydrolysis by NUDT9-H from *Salpingoeca rosetta* (srNUDT9-H) as a function of [ADPR], measured at pH 8.5 in the presence of 16 mM $Mg^{2+}$, normalized to the rate measured at the highest ADPR concentration (see Materials and methods for details). *The solid line* is a fit to the Michaelis-Menten equation, the $K_M$ value is indicated. Data are shown as mean ± SEM of 3 experiments. (**B**) $Mg^{2+}$-dependence of $k_{cat}$ (s$^{-1}$) for srNUDT9-H, determined at fixed pH values of 8.5 (*black symbols*) and 7.1 (*green symbols*) in the presence of saturating ADPR. *Solid lines* are fits to the equation

$$k_{cat} = \left(k_{cat1} \cdot K_2^{n_2} \cdot [Mg^{2+}]^{n_1} + k_{cat2} \cdot [Mg^{2+}]^{n_1+n_2}\right) / \left(K_1^{n_1} \cdot K_2^{n_2} + K_2^{n_2} \cdot [Mg^{2+}]^{n_1} + [Mg^{2+}]^{n_1+n_2}\right),$$ yielding fit parameters of $k_{cat1}$ = 1.57 ± 0.08 s$^{-1}$, $K_1$ = 0.031 ± 0.002 mM, $n_1$ = 2.8 ± 0.9, $k_{cat2}$ = 3.59 ± 0.19 s$^{-1}$, $K_2$ = 29 ± 8 mM, $n_2$ = 0.86 ± 0.18 (pH 8.5, *black*), and $k_{cat1}$ = 1.76 ± 0.07 s$^{-1}$, $K_1$ = 0.102 ± 0.008 mM, $n_1$ = 1.8 ± 0.2, $k_{cat2}$ = 3.44 ± 0.12 s$^{-1}$, $K_2$ = 35 ± 6 mM, $n_2$ = 1.0 ± 0.2 (pH 7.1, *green*). (**C**) Estimated $k_{cat}$ values (s$^{-1}$) for full-length srTRPM2 (*brown*, calculated per subunit of protein) and srNUDT9-H (*black*), determined in the presence of saturating ADPR at pH 8.5 in presence of 16 mM $Mg^{2+}$. Data are shown as mean ± SEM of at least 3 experiments. See also *Figure 3—figure supplement 1*.

DOI: https://doi.org/10.7554/eLife.44556.008

The following source data and figure supplement are available for figure 3:

**Source data 1.** Enzymatic activity of full-length srTRPM2 and of the isolated srNUDT9-H domain.
DOI: https://doi.org/10.7554/eLife.44556.010
**Figure supplement 1.** Purification of srNUDT9-H.
DOI: https://doi.org/10.7554/eLife.44556.009

bound $Mg^{2+}$ ions. In contrast, $Mg^{2+}$ binding to the low-affinity site was insensitive to pH ($K_{1/2}$~30 mM), suggesting that this site is further away from the catalytically important titratable side chain.

We also expressed full-length srTRPM2 in HEK-293S cells, and attempted to purify the detergent-solubilized protein. However, in contrast to full-length nvTRPM2, the size-exclusion chromatogram of srTRPM2 suggested that the protein was not monodisperse, and repeated chromatography of the freshly isolated peak fraction already showed signs of aggregation (*Figure 3—figure supplement 1A*). Nevertheless, tentative ADPRase assays using freshly isolated full-length srTRPM2 revealed a $k_{cat}$ value (lower estimate) of ~1.6 $s^{-1}$/subunit (*Figure 3C*, *brown bar*), comparable to that obtained for isolated srNUDT9-H (*Figure 3C*, *black bar*) under identical conditions (pH 8.5, 16 mM $[Mg^{2+}]$). Thus, despite species-specific differences in its precise catalytic mechanism, ADPRase activity seems to be a shared feature of invertebrate TRPM2 proteins.

## Disrupting catalysis at the NUDT9-H domain does not affect macroscopic gating parameters of nvTRPM2

The small group of known channel-enzyme proteins, in which a single polypeptide chain forms both a transmembrane ion pore and a catalytically active domain, includes TRPM6, TRPM7, and the Cystic Fibrosis Transmembrane Conductance Regulator (CFTR) chloride ion channel. In CFTR, ATP hydrolysis cycles at cytosolic nucleotide-binding domains are strictly coupled to conformational changes that open and close (gate) the channel pore (*Csanády et al., 2010*). In contrast, the catalytic activity of the cytosolic kinase domain in TRPM6/7 is not linked to gating conformational changes of those channels (*Krapivinsky et al., 2014*). Comparison of the ligand-free, closed, and ADPR+$Ca^{2+}$-bound, open, conformations of zebrafish and human TRPM2 suggested large movements of the four NUDT9-H domains upon opening of ADPR-bound channels. These rearrangements appear to stabilize the open state (*Huang et al., 2018*; *Wang et al., 2018*), suggesting a potential influence of NUDT9-H ligand-occupancy on channel gating. To test whether any coupling exists between the catalytic cycle at the nvNUDT9-H domain and nvTRPM2 channel gating, we employed three independent strategies to inhibit hydrolysis of the bound ligand, and compared their impacts on nvNUDT9-H ADPRase activity and on the activity of full-length nvTRPM2 channels.

Consistent with the exquisite $Mg^{2+}$ dependence of nvNUDT9-H catalysis (*Figure 2C*), omission of $Mg^{2+}$ from the reaction buffer with or without additon of 100 μM CDTA (free $[Mg^{2+}]$ low nanomolar and low micromolar, respectively) abolished ADPRase activity of the protein to below the limit of detection of our assay (*Figure 4A*). Indeed, although no significant contamination by inorganic phosphate ($P_i$) could be detected in any of the compounds used in the assay (*Figure 4A*, *bars 1–4*), $P_i$ was released at a low but detectable rate by alkaline phosphatase (AP) in the presence of ADPR (*Figure 4A*, *bars 5–7*). This background signal, which is independent of free $[Mg^{2+}]$ (compare three *purple bars*), reflects slow spontaneous hydrolysis of ADPR at alkaline pH (*Iordanov et al., 2016*). In contrast, the robust ADPRase activity of nvNUDT9-H measured in millimolar free $[Mg^{2+}]$ (*Figure 4A*, *green bar*) was reduced to this background level when free $[Mg^{2+}]$ was lowered to micro- or nanomolar (*Figure 4A*, compare *orange bars* to *purple bars*). Control experiments using AMP as a substrate confirmed that the activity of the co-applied alkaline phosphatase was independent of free $[Mg^{2+}]$ and not rate limiting for the assay (*Figure 4A*, *white bars*). We next tested the effect of cytosolic free $[Mg^{2+}]$ on macroscopic nvTRPM2 currents, in inside-out patches excised from *Xenopus laevis* oocytes expressing nvTRPM2 (*Figure 4B*). Currents were repeatedly activated by cytosolic exposure to $Ca^{2+}$ plus ADPR (*black and purple bars*), either in the presence of 2 mM cytosolic $Mg^{2+}$ (*green bars*), or in the presence of 100 μM CDTA (*orange bar*). Interestingly, currents activated in the absence of cytosolic $Mg^{2+}$ were almost twofold larger (*Figure 4B*; *Figure 4C*, *left pair of bars*). However, this effect reflected a similarfold increase in unitary current amplitude (*Figure 4C*, *right pair of bars*), from ~−2.5 pA to ~−4 pA at −20 mV (*Figure 4—figure supplement 1*), rather than a change in channel open probability: in the presence of cytosolic $Mg^{2+}$, at −20 mV membrane potential, $Na^+$ influx through open nvTRPM2 channel pores is blocked by cytosolic $Mg^{2+}$ ions. Moreover, the time constants of channel deactivation upon sudden removal of cytosolic ADPR (*Figure 4B*, *colored numbers*, in ms), obtained from single-exponential fits (*colored lines*) was not measurably prolonged by $Mg^{2+}$ removal (*Figure 4D*), as would be expected if ADPR hydrolysis facilitated pore closure. Thus, while removal of cytosolic $Mg^{2+}$ disrupts catalytic activity at the nvNUDT9-H domain, it does not affect macroscopic gating parameters of nvTRPM2 channels.

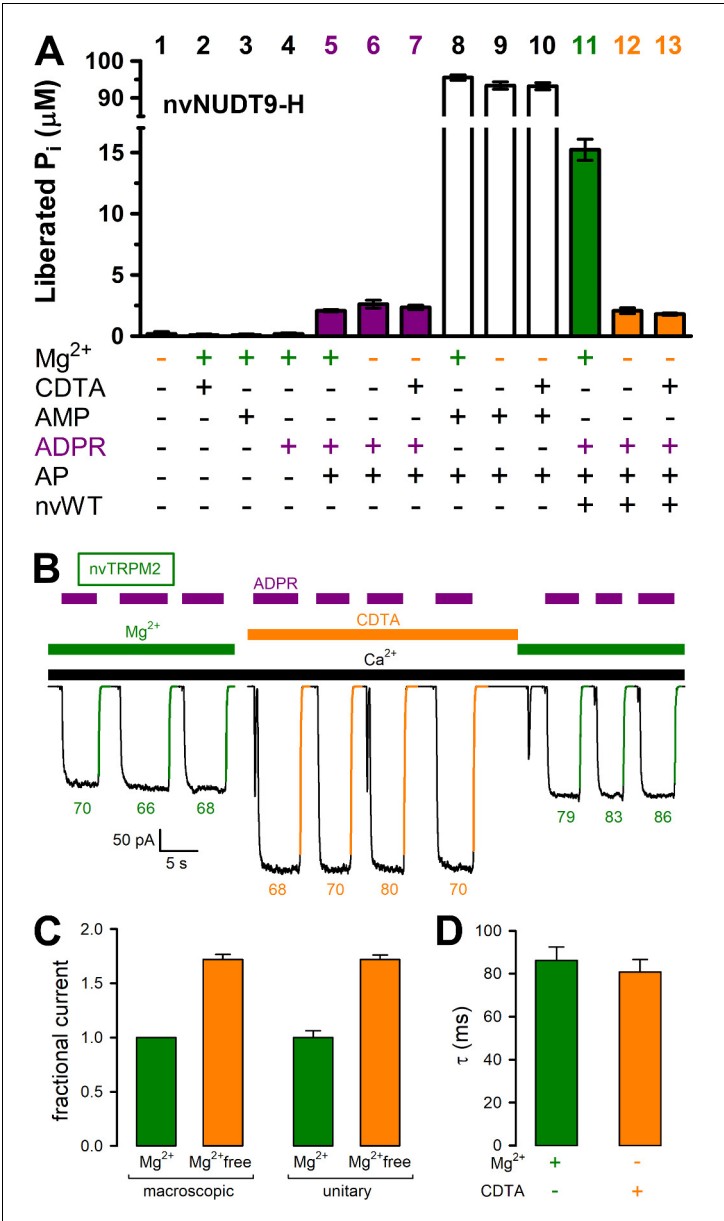

**Figure 4.** Mg$^{2+}$ removal abolishes hydrolytic activity but does not affect macroscopic gating parameters of nvTRPM2. (**A**) ADPRase rates of WT nvNUDT9-H (nvWT), reported as P$_i$ (in μM) released by co-applied alkaline phosphatase (AP), in sample mixtures containing, as indicated, MgCl$_2$ (16 mM), CDTA (100 μM), AMP (100 μM), ADPR (100 μM), AP (2–3 U), nvNUDT9-H (0.5 nM), and incubated for 5 min at room temperature. Data are shown as mean ± SEM of 3 experiments. (**B**) Macroscopic inward currents at −20 mV membrane potential, activated by repeated exposures to 100 μM ADPR (*purple bars*) in the presence of 125 μM Ca$^{2+}$ (*black bar*), in an inside-out patch excised from a *Xenopus laevis* oocyte injected with nvTRPM2 cRNA. Cytosolic (bath) solution contained either 2 mM added Mg(gluconate)$_2$ (*green bars*) or no added Mg$^{2+}$ but 100 μM CDTA (*orange bar*). *Colored solid lines are exponentials fitted to the decay time courses, with time constants (in ms) indicated.* (**C**) Fractional changes in macroscopic (*left*) and unitary (*right*) current amplitudes upon removal of cytosolic Mg$^{2+}$ at −20 mV membrane potential. Macroscopic currents in the absence of cytosolic Mg$^{2+}$ were normalized to those in the presence of 2 mM cytosolic Mg$^{2+}$ within the same patch, *left orange bar* shows mean ± SEM from 11 patches. Average unitary current in the absence (*right orange bar*; mean ± SEM from 4 patches) and presence (*right green bar*; mean ± SEM from 3 patches) of cytosolic (2 mM) Mg$^{2+}$ are shown normalized to the mean of the latter value (−2.5 pA). (**D**) Average time constants (mean ± SEM from 11 patches) of macroscopic current decay following ADPR removal in the presence (*green*) or absence (*orange*) of cytosolic Mg$^{2+}$. See also *Figure 4—figure supplement 1*, *Figure 4—figure supplement 2*, and *Figure 4—figure supplement 3*.

*Figure 4 continued on next page*

*Figure 4 continued*

DOI: https://doi.org/10.7554/eLife.44556.011

The following source data and figure supplements are available for figure 4:

**Source data 1.** Mg$^{2+}$ removal abolishes hydrolytic activity but does not affect macroscopic gating parameters of nvTRPM2.

DOI: https://doi.org/10.7554/eLife.44556.015

**Figure supplement 1.** Effects of extra- and intracellular Mg$^{2+}$ on unitary conductance properties of nvTRPM2.

DOI: https://doi.org/10.7554/eLife.44556.012

**Figure supplement 2.** nvTRPM2 channels are permeable to Mg$^{2+}$.

DOI: https://doi.org/10.7554/eLife.44556.013

**Figure supplement 3.** Lack of background currents in water-injected *Xenopus laevis* oocytes in symmetrical Na-gluconate solutions.

DOI: https://doi.org/10.7554/eLife.44556.014

The ADPR analog α-β-methylene-ADPR (AMPCPR) (*Pankiewicz et al., 1997*) was shown to be resistant to hydrolysis by several Nudix enzymes including human NUDT5 (*Zha et al., 2008*) and NUDT9 (*Tóth et al., 2014*), but supports channel activity of human TRPM2 channels (*Tóth et al., 2014*). Correspondingly, whereas the slow spontaneous hydrolysis of ADPR was accelerated by nvNUDT9-H to an extent roughly proportional to the concentration of the enzyme (*Figure 5A*, *bars 2, 5*, and *7*), AMPCPR showed no sign of spontaneous hydrolysis, and remained unhydrolyzed in the presence of increasing amounts of nvNUDT9-H protein (*Figure 5A*, *bars 3, 6*, and *8*). On the other hand, AMPCPR readily activated nvTRPM2 channel currents, although, compared to ADPR, higher (tens of micromolar) concentrations of the analog were required. Moreover, even in the presence of a quasi-saturating concentration of AMPCPR, currents remained smaller than in 100 μM ADPR (*Figure 5B–C*). Upon nucleotide removal, AMPCPR-activated nvTRPM2 currents declined ~1.5 times faster than ADPR-activated currents (*Figure 5B*, *colored fit lines* and *time constants*; *Figure 5D*, p = 0.02), suggesting a somewhat destabilized open state for AMPCPR-bound channels. Thus, AMPCPR is a partial agonist for nvTRPM2, just as for human TRPM2 (*Tóth et al., 2014*). Nevertheless, it clearly supports pore gating in the complete absence of nvNUDT9-H catalytic activity.

To disrupt catalysis through mutagenesis, we introduced the double mutations E1443I/F1444L (Nudix box: AILGEE) and E1446K/E1447K (Nudix box: AEFGKK) into the nvNUDT9-H domain. The mutant proteins were expressed at similar amounts as wild-type (WT) nvNUDT9-H, and remained similarly monodisperse in solution (*Figure 2—figure supplement 1B*), confirming proper folding of the mutant domains. In ADPRase activity assays neither double mutation was found to greatly impair the affinity for ADPR binding, as reflected by K$_M$ values which remained within twofold of WT (*Figure 6A*). However, maximal turnover rate was dramatically impaired by both double mutations: k$_{cat}$ was ~1% of WT for E1443I/F1444L (*Figure 6B*, *red bar*), and ~0.1% of WT for E1446K/E1447K (*Figure 6B*, *blue bar*), consistent with the reported effects of the analogous mutations on the catalytic activity of human NUDT9 (*Perraud et al., 2003*). Whereas enzymatic activity of nvNUDT9-H was nearly abolished by both double mutations, full-length nvTRPM2 channels bearing the same double mutations generated macroscopic currents that were activated by low micromolar concentrations of ADPR (*Figure 6D–E*), just as WT nvTRPM2 (*Figure 6C*). Neither the apparent affinity for macroscopic current activation by ADPR (K$_{1/2}$ ~2 μM; *Figure 6F*), nor the time constant of current deactivation upon ADPR removal (τ ~100 ms; *Figure 6C–E*, *colored lines* and *time constants*; *Figure 6G*) were significantly affected by either double mutation (p > 0.17).

## Catalytic cycle at the NUDT9-H domain is not coupled to gating conformational changes

The above studies on macroscopic currents reveal that Mg$^{2+}$ removal has little effect on channel open probability (compare *Figure 4B* and *Figure 4—figure supplement 1A*), and that the two double mutations do not impair the apparent affinity for current activation by ADPR. On the other hand, these data do not provide information on potential changes in steady-state gating kinetics of single nvTRPM2 channels. For instance, any effects of the mutations on channel opening rate, or parallel changes in opening and closing rate upon Mg$^{2+}$ removal, would remain undetected in the macroscopic recordings. We therefore extracted steady-state single-channel gating transition rates in the

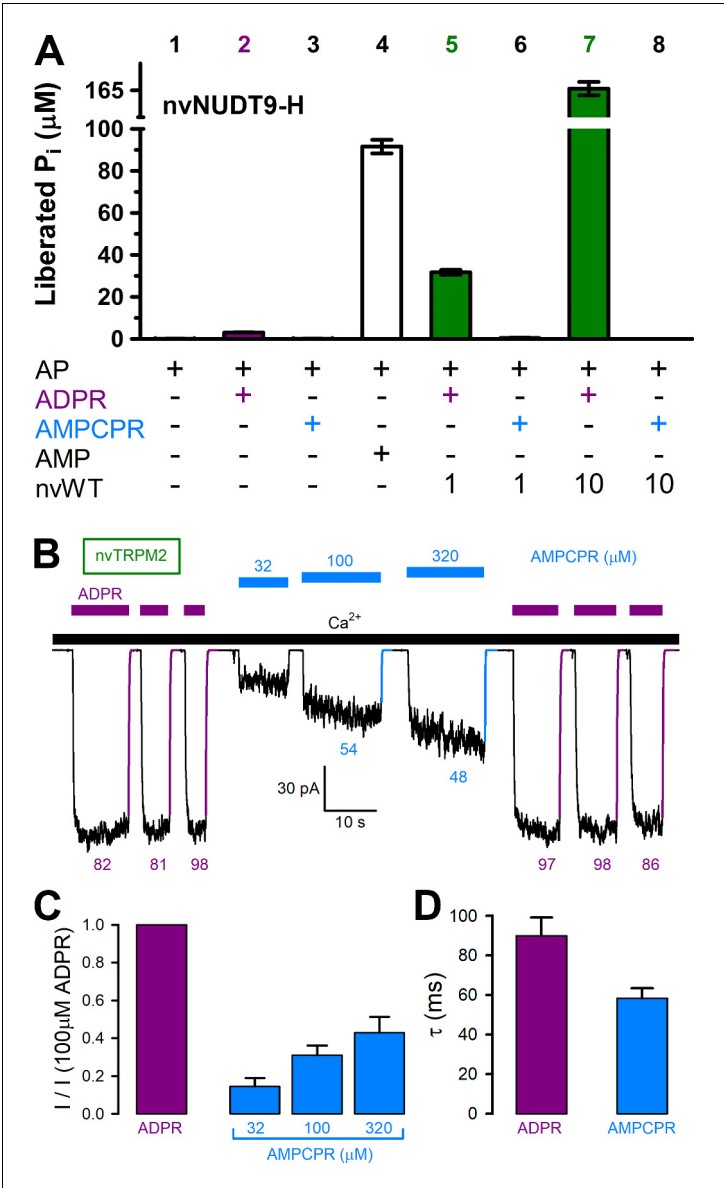

**Figure 5.** AMPCPR cannot be hydrolyzed by nvTRPM2, but still activates the channel. (**A**) Nucleoside diphosphohydrolase rates of WT nvNUDT9-H (nvWT), reported as $P_i$ (in µM) released by co-applied alkaline phosphatase (AP), in sample mixtures containing, as indicated, AP (2–3 U), ADPR (100 µM), AMPCPR (100 µM), AMP (100 µM), nvNUDT9-H (1 or 10 nM), and incubated for 5 min at room temperature in the presence of 16 mM $Mg^{2+}$, pH=8.5. Data are shown as mean ± SEM of 3 experiments. (**B**) Macroscopic nvTRPM2 currents activated by repeated exposures to (100 µM) ADPR (*purple bars*) or various concentrations of AMPCPR (*blue bars*) in the presence of 125 µM $Ca^{2+}$ (*black bar*). *Colored solid lines* are fitted exponentials with time constants (in ms) indicated. (**C**) Fractional current activation by indicated concentrations of AMPCPR (*blue bars*; mean ± SEM from 4 to 7 patches), normalized to the current elicited in the same patch by 100 µM ADPR (*purple bar*). (**D**) Average macroscopic current decay time constants (mean ± SEM from 5 patches) following removal of the activating nucleotide.

DOI: https://doi.org/10.7554/eLife.44556.016

The following source data is available for figure 5:

**Source data 1.** AMPCPR cannot be hydrolyzed by nvTRPM2, but still activates the channel.
DOI: https://doi.org/10.7554/eLife.44556.017

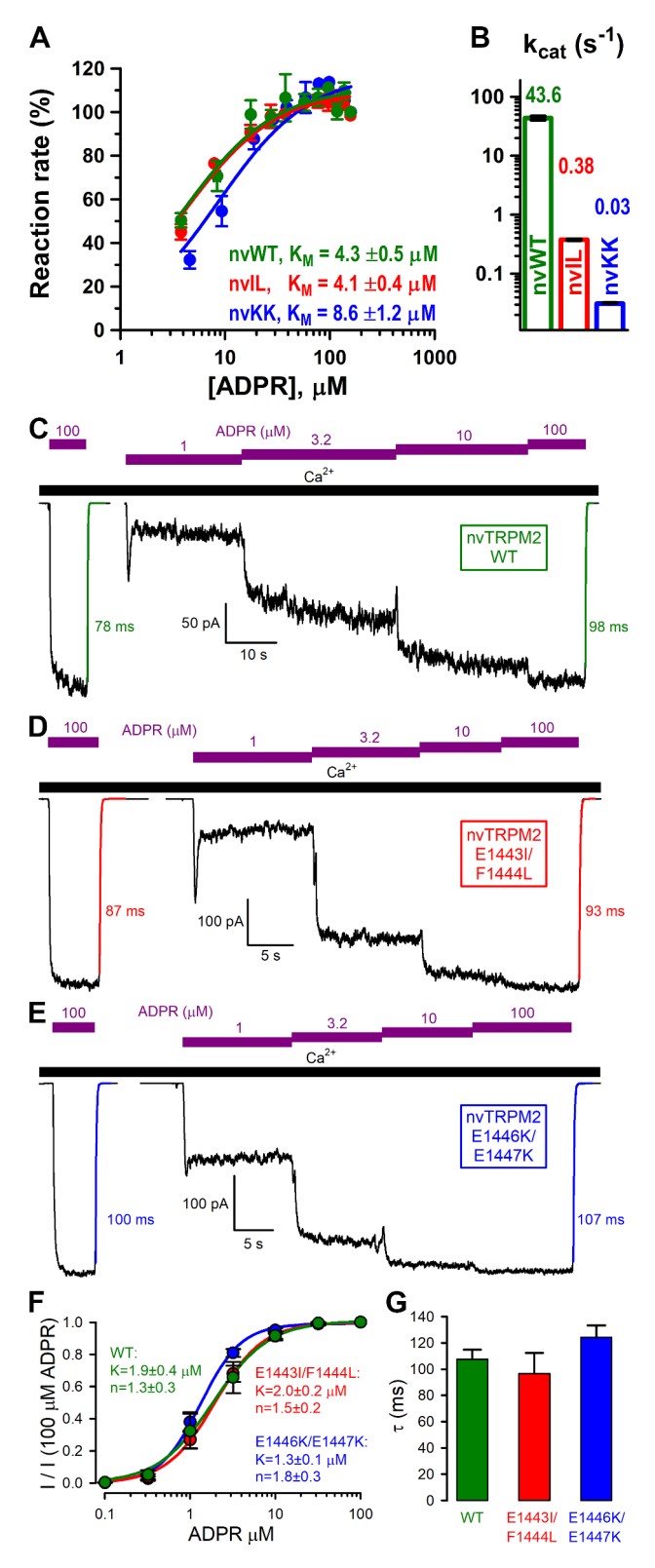

**Figure 6.** Nudix box substitutions impair catalytic activity, but not macroscopic gating properties, of nvTRPM2. (**A**) Rates of ADPR hydrolysis by nvNUDT9-H WT (*green*), E1443I/F1444L (nvIL, *red*), and E1446K/E1447K (nvKK, *blue*) in 16 mM Mg$^{2+}$, pH=8.5, as a function of [ADPR], normalized to the rate measured at 160 µM ADPR. *Solid lines* are fits to the Michaelis-Menten equation, K$_M$ values are indicated. (**B**) Estimated k$_{cat}$ values (s$^{-1}$) for nvNUDT9-H WT
*Figure 6 continued on next page*

*Figure 6 continued*

(*green*), E1443I/F1444L (*nvIL*, *red*), E1446K/E1447K (*nvKK*, *blue*), estimated in the presence of saturating ADPR. Data are shown as mean ± SEM of at least 3 experiments. (C–E) Macroscopic currents of (C) WT, (D) E1443I/F1444L, and (E) E1446K/E1447K nvTRPM2 activated by exposures to various concentrations of ADPR (*purple bars*) in the presence of 125 µM Ca$^{2+}$ (*black bar*). *Colored solid lines* are fitted exponentials with time constants (in ms) indicated. (F) Fractional current activation as a function of [ADPR] for WT (*green*), E1443I/F1444L (*red*), and E1446K/E1447K (*blue*) nvTRPM2, normalized to the current in 100 µM ADPR in the same patch. *Symbols* and *error bars* represent mean ± SEM (n = 3–6), *solid lines* are fits to the Hill equation with fit parameters indicated. (G) Average macroscopic current decay time constants (mean ± SEM from 6 patches) following ADPR removal for WT and mutant nvTRPM2.

DOI: https://doi.org/10.7554/eLife.44556.018

The following source data and figure supplement are available for figure 6:

**Source data 1.** Nudix box substitutions impair catalytic activity, but not macroscopic gating properties, of nvTRPM2.
DOI: https://doi.org/10.7554/eLife.44556.020

**Figure supplement 1.** Sensitivity of the ADPRase assay allows reliable detection of enzymatic activity for mutant nvNUDT9-H and srNUDT9-H proteins.
DOI: https://doi.org/10.7554/eLife.44556.019

presence of saturating (100 µM) ADPR for WT nvTRPM2 in the presence and absence of cytosolic Mg$^{2+}$ (*Figure 7A*), as well as for E1443I/F1444L (*Figure 7B*) and E1446K/E1447K (*Figure 7C*) nvTRPM2 channels in the presence of Mg$^{2+}$, by studying microscopic currents in patches containing 1–12 active channels in which individual gating transitions remained well resolved (*Figure 7A–C*, *yellow insets* with expanded time scale). As long as the number of active channels in the patch can be estimated with confidence, the gating rates of individual channels can be reliably extracted from such recordings by maximum likelihood fits to the dwell-time distributions (see Materials and methods). Dwell-times at the highest conductance level (i.e. with all channels open) were single-exponentially distributed, whereas at all other conductance levels the dwell-time distributions showed two exponential components (*Figure 7—figure supplement 1*). Such a pattern of dwell-time histograms uniquely identifies a bursting gating pattern with one open and two closed states, as shown earlier for hsTRPM2 (*Csanády and Törocsik, 2009*; *Tóth et al., 2014*), which displays two distinct populations of closed events (long, >100 ms, 'interburst'; brief,~2 ms, 'flickery') but a single population of open events. Using a three-state C$_{slow}$↔O↔C$_{fast}$ scheme (*Tóth et al., 2014*) to model nvTRPM2 gating, we extracted unitary transition rates (*Figure 7—figure supplement 1*) and calculated open probabilities (P$_o$; *Figure 7D*), mean open burst durations (τ$_b$; *Figure 7E*), and mean closed interburst durations (τ$_{ib}$; *Figure 7F*). As expected from the macroscopic current measurements (*Figure 4B–C*, cf., *Figure 4—figure supplement 1A*), Mg$^{2+}$ removal did not affect open probability of WT nvTRPM2 (*Figure 7D*, *orange* vs. *green* bar), although it slightly accelerated channel gating: both mean burst and interburst durations became ~30% shorter (*Figure 7E–F*, *orange* vs. *green* bars). Moreover, for both double mutants, open probabilities and mean burst durations remained similar to those of WT, and mean interburst durations were altered by less than twofold (*Figure 7D–F*, *red* and *blue* bars vs. *green* bar). All the subtle effects on single-channel gating parameters caused by the mutations or Mg$^{2+}$ removal (*Figure 7D–F*) proved statistically insignificant (p > 0.18).

## Invertebrate TRPM2 pores are stable, whereas vertebrate TRPM2 channels inactivate

A key characteristic feature of human TRPM2 (hsTRPM2) is a rapid inactivation process, even in the presence of its activating ligands (*Csanády and Törocsik, 2009*; *Tóth and Csanády, 2012*), which is not observed for nvTRPM2 (*Zhang et al., 2018*). Inactivation of hsTRPM2 reflects a conformational change of the pore (*Tóth and Csanády, 2012*), and is linked specifically to the amino acid sequence of its short three-residue post-filter helix which lines the extracellular pore vestibule: indeed, swapping the corresponding residues between nvTRPM2 and hsTRPM2 confers inactivation to the anemone channel but eliminates it from the human channel (*Zhang et al., 2018*). Comparison of the structures of nvTRPM2 (*Zhang et al., 2018*) and hsTRPM2 (*Wang et al., 2018*) (*Figure 8A*, *salmon* and *cyan*) indeed reveal marked differences between the two proteins in this region. In the external pore vestibule of nvTRPM2, the acidic side chains of D1041 and E1042 in the post-filter helix, and of

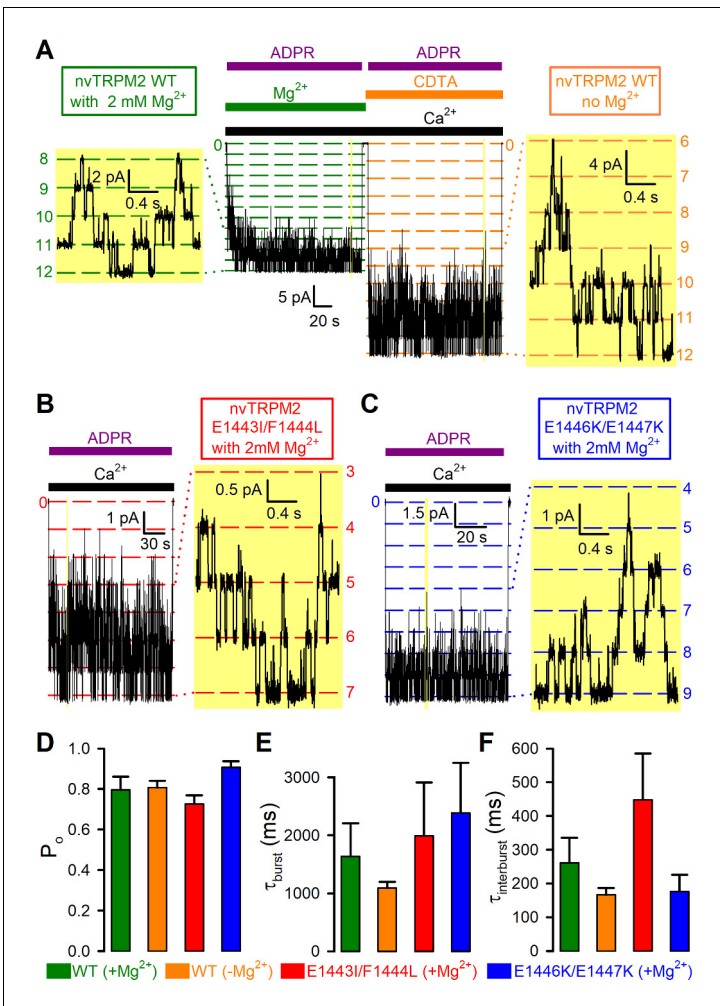

**Figure 7.** Mg$^{2+}$ removal and Nudix box mutations little affect steady-state single-channel gating kinetics of nvTRPM2 channels. (A–C) Steady-state channel currents from patches containing (A) twelve WT (B) seven E1443I/F1444L and (C) nine E1446K/E1447K nvTRPM2 channels, activated by exposure to 125 µM Ca$^{2+}$ and 100 µM ADPR (*bars*) in the presence of 2 mM cytosolic Mg$^{2+}$; channel number was estimated in each patch as the maximum number of simultaneously open channels (see Materials and methods). In (A), WT channels were reopened by a second application of ADPR in the absence of bath Mg$^{2+}$ (*orange bar*: Mg$^{2+}$ removal + addition of 100 µM CDTA). *Narrow yellow boxes* highlight segments of record shown to the left/right at expanded time/current scales; note well resolved gating transitions. Bandwidth, 200 Hz. (D–F) Open probabilities (D), mean burst (E) and interburst (F) durations obtained from multi-channel fits (see *Figure 7—figure supplement 1*), for WT nvTRPM2 with (*green*) or without (*orange*) Mg$^{2+}$, and for E1443I/F1444L (*red*) and E1446K/E1447K (*blue*) nvTRPM2 with Mg$^{2+}$. Data are shown as mean ± SEM from 6 to 12 patches. See also *Figure 7—figure supplement 1*.
DOI: https://doi.org/10.7554/eLife.44556.021

The following source data and figure supplement are available for figure 7:

**Source data 1.** Mg$^{2+}$ removal and Nudix box mutations little affect steady-state single-channel gating kinetics of nvTRPM2 channels.
DOI: https://doi.org/10.7554/eLife.44556.023

**Figure supplement 1.** Kinetic analysis of multichannel patches by a simultaneous maximum likelihood fit to the dwell-time histograms of all conductance levels.
DOI: https://doi.org/10.7554/eLife.44556.022

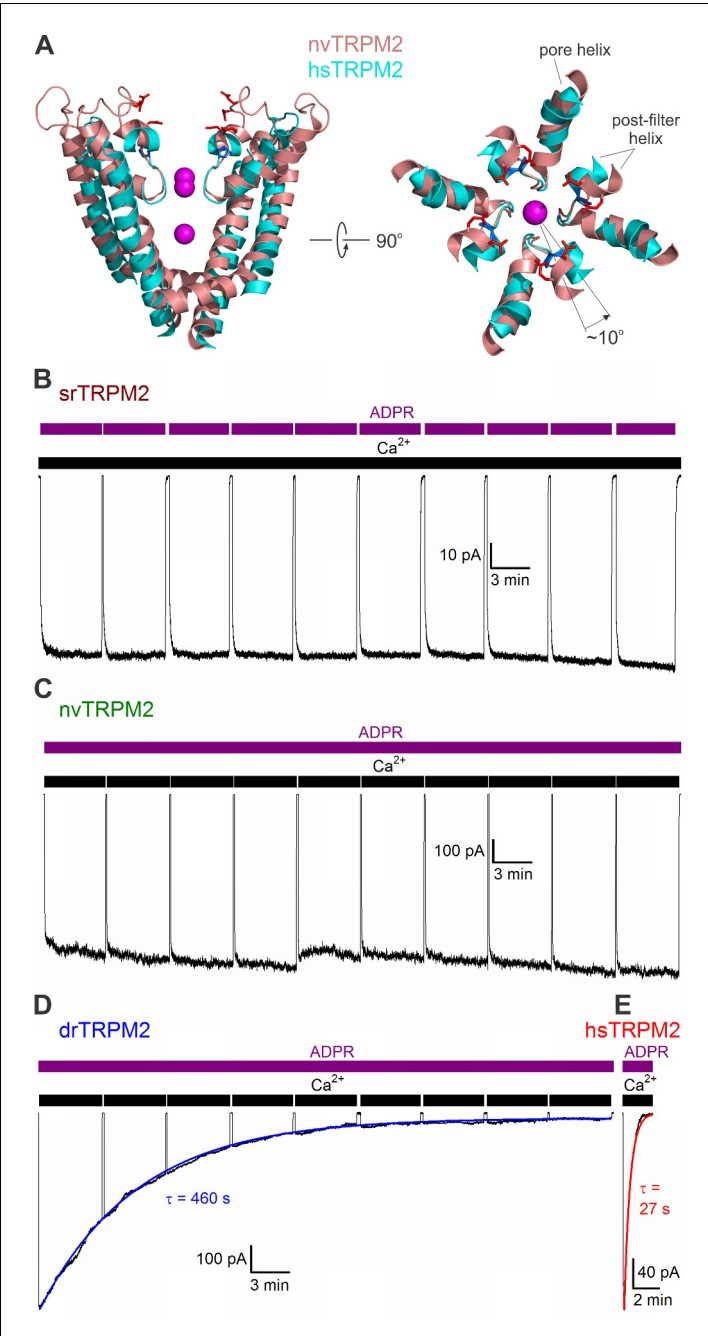

**Figure 8.** Pore structure and inactivation properties of invertebrate and vertebrate TRPM2 channels. (**A**) Superposition of nvTRPM2 (*salmon*, PDBID: 6CO7) and hsTRPM2 (*cyan*, PDBID: 6MIX) pore structures, viewed from an angle parallel to (*left*, front and rear subunit removed), or perpendicular to (*right*, only pore helices, filters, and post-filter helices are shown), the membrane plane. Na$^+$ ions in the nvTRPM2 structure are shown as *magenta spheres*, nvTRPM2 residues D1041, E1042, E1046, and E1050 (*red*), and hsTRPM2 residue P983 (*blue*) are shown as sticks. (**B–E**) Macroscopic currents of (**B**) srTRPM2, (**C**) nvTRPM2, (**D**) drTRPM2, and (**E**) hsTRPM2 channels activated by prolonged exposure to 100 µM ADPR (*purple bars*) plus 125 µM Ca$^{2+}$ (*black bars*). In (**B–D**) Ca$^{2+}$ or ADPR was briefly removed every ~5 min to verify seal integrity. *Blue and red lines* in (**D–E**) are fitted exponentials with time constants (in s) indicated.

DOI: https://doi.org/10.7554/eLife.44556.024

The following source data is available for figure 8:

**Source data 1.** Pore structure and inactivation properties of invertebrate and vertebrate TRPM2 channels.
DOI: https://doi.org/10.7554/eLife.44556.025

E1046 and E1050 in the post-filter loop, form a double ring of negative charges (*Figure 8A*, *red sticks*). Not only are those charges absent in hsTRPM2, but when the pore helices and pore loops of the two structures are aligned, the axis of the post-filter helix of hsTRPM2 is rotated by ~10° (counterclockwise, when viewed from the extracellular side) relative to that of nvTRPM2 (*Figure 8A*, *right*). This displacement of the hsTRPM2 post-filter helix is caused by the appearance of a proline (P983) at its N-terminal end (*Figure 8A*, *blue sticks*).

Interestingly, a sequence alignment of the pore regions of TRPM2 orthologs (*Figure 1*, *left*) reveals that these changes in pore sequence, responsible for inactivation of the human channel, also appeared between chordates and vertebrates: the residue triplet which forms the negatively charged post-filter helix in nvTRPM2 is intact (*Figure 1*, Post-filter helix, *blue sequence motifs*) and preceded by a phenylalanine in all invertebrates, but is replaced by an uncharged doublet (*Figure 1*, Post-filter helix, *red sequence motifs*) and preceded by a proline in vertebrates. Furthermore, the glutamate side chain in the selectivity filter responsible for the very high $Ca^{2+}$ preference of the nvTRPM2 pore (*Zhang et al., 2018*) was also replaced by an uncharged side chain in vertebrates (*Figure 1*, Filter, *blue vs. red residues*), lowering $Ca^{2+}$ permeability.

To address whether a stable pore is a general feature of invertebrate TRPM2 channels and inactivation a general feature of vertebrate channels, we sought to test this feature for an additional member in both groups. To this end, we expressed full-length srTRPM2 (invertebrate) and drTRPM2 (vertebrate) channels in *Xenopus laevis* oocytes, and studied their functional properties in excised inside-out patches. Expression of both proteins caused the appearance of large macroscopic currents that depended on the presence of cytosolic $Ca^{2+}$ and ADPR (*Figure 8B,D*). Currents generated by srTRPM2 channels (*Figure 8B*) remained stable for over the time course of an hour, just like nvTRPM2 currents (*Figure 8C*). In contrast, drTRPM2 currents inactivated in the maintained presence of ADPR+$Ca^{2+}$ (*Figure 8D*) just like those of hsTRPM2 (*Figure 8E*), although for the zebrafish channel the rate of inactivation was an order of magnitude slower (*Figure 8D–E*, compare time constants of single-exponential fits, average τ was 655 ± 178 s (n = 12) for drTRPM2, and 29 ± 2 s (n = 4) for hsTRPM2).

## Discussion

Based on sequence homology of its C-terminal NUDT9-H domain to the mitochondrial ADPRase NUDT9, the human TRPM2 channel was suggested to act as a channel-enzyme (*Perraud et al., 2001*), but was later found catalytically inactive (*Iordanov et al., 2016*). For nvTRPM2, enzymatic activity has been suggested based on indirect evidence (*Kühn et al., 2016*). Here, we show that the ancient TRPM2 channels of the choanoflagellate *Salpingoeca rosetta* and of the sea anemone *Nematostella vectensis* are indeed true chanzymes which, unlike their human ortholog, hydrolyze their activating ligand ADPR (*Figures 2–3*). This is consistent with the presence of canonical Nudix box sequences in all invertebrate TRPM2 channels (*Figure 1*, *right*, *blue sequences*). We could not ascertain whether the mutations in the key Nudix box positions (*Figure 1*, *right*, *asterisks*) are solely responsible for the loss of catalysis in vertebrate TRPM2 channels, as we were unable to obtain hsNUDT9-H constructs with 'revertant' Nudix box motifs (Nudix box: REFRQE or REFGEE) in soluble form. Thus, although the Nudix box mutations are clearly *sufficient* to abrogate enzymatic activity (*Figure 6B*, *Perraud et al., 2003*), it remains possible that in vertebrate TRPM2 channels additional mutations have accumulated which are also incompatible with catalysis (cf., *Kühn et al., 2015*).

For both srTRPM2 and nvTRPM2, the isolated NUDT9-H domains serve as convenient soluble model systems with $k_{cat}$ values identical to those of the full-length proteins (*Figures 2B* and *3C*). We observed a somewhat (~3 fold) larger $K_M$ value for full-length nvTRPM2 compared to its isolated NUDT9-H domain (*Figure 2A*), a slight discrepancy which might be explained by structural constraints imposed on the nvNUDT9-H domain by the rest of the channel protein, which is expected to reside in a closed-channel conformation under the conditions of our enzymatic assay (in the absence of $PIP_2$ and $Ca^{2+}$).

The steep $Mg^{2+}$ dependence (Hill coefficient ~2, *Figure 2C*) of the molecular turnover rate for nvTRPM2 is consistent with at least two $Mg^{2+}$ ions coordinated in the vicinity of the bound nucleotide, and its pH-dependence (*Figure 2D*) with general base catalysis, as found in other members of the Nudix hydrolase family (*Mildvan et al., 2005*). Of note, whereas in all Nudix enzymes an intact Nudix box motif is required for structural integrity of the active site and for $Mg^{2+}$ coordination, the

identity of the actual catalytic base has greatly diverged. In some members, it is provided by a Nudix-box glutamate (pK$_a$ ~7.6; *Harris et al., 2000*), in others by a glutamate (*Gabelli et al., 2002*) or a histidine (pKa ~6.7; *Legler et al., 2002*) outside the Nudix box, and in the closely related NUDT9 enzyme the identity of the catalytic base is unknown (*Shen et al., 2003*). For nvTRPM2, the observed pK$_a$ of ~6.8 in high Mg$^{2+}$ (*Figure 2D*) would be consistent with either scenario.

For srTRPM2, the molecular turnover rate was smaller than for nvTRPM2 (compare *Figure 3C* to *Figure 2B*), but clearly measurable. Indeed, our sensitive enzymatic assay allows reliable quantitation of $k_{cat}$ values as low as 0.04 s$^{-1}$: even for such slow enzymes, employing higher protein concentrations allows robust separation of the enzymatic activity from the background signal caused by spontaneous ADPR hydrolysis (*Figure 6—figure supplement 1*). In addition to a Mg$^{2+}$ and pH dependence qualitatively similar to that observed for nvNUDT9-H, srNUDT9-H catalytic activity was further augmented at Mg$^{2+}$ concentrations in the tens of millimolar range (*Figure 3B*). The physiological relevance of such a low-affinity Mg$^{2+}$ binding site is unclear, but might potentially become important, considering that these marine choanoflagellates live in sea water which contains ~50 mM Mg$^{2+}$. Although we did not test Mg$^{2+}$ permeability for srTRPM2, both hsTRPM2 (*Tóth and Csanády, 2012*) and nvTRPM2 (*Figure 4—figure supplement 2*) channels are highly permeable to Mg$^{2+}$. Thus, in marine invertebrate organisms, under physiological conditions, local cytosolic Mg$^{2+}$ concentrations might become elevated in the vicinity of an open TRPM2 channel pore.

In some chanzymes, such as the CFTR channel, the catalytic cycle is strictly coupled to gating conformational changes (*Csanády et al., 2010*), whereas in others, such as TRPM6/7, the two processes occur entirely uncoupled from each other (*Krapivinsky et al., 2014*). The recent discovery of an additional ligand-binding site in TRPM2, implied by intact ADPR-dependent currents of nvTRPM2 channels lacking the NUDT9-H domain (*Kühn et al., 2016*) and confirmed in the structure of zebrafish TRPM2 (*Huang et al., 2018*), rendered *strict* coupling between enzymatic activity and channel gating unlikely. On the other hand, comparison of closed (apo) and open (ADPR-bound) structures of both zebrafish and human TRPM2 revealed large gating-associated conformational changes in the NUDT9-H domains that seemed to contribute to open-state stability (*Huang et al., 2018*; *Wang et al., 2018*). It thus remained a possibility that ADPR binding to NUDT9-H, and hence the catalytic cycle of enzymatically active ancient orthologs, might impact on channel gating (*loose coupling*).

Using three independent approaches, we show here that complete (or near-complete) disruption of catalysis at the nvNUDT9-H domain fails to affect macroscopic or microscopic gating parameters of full-length nvTRPM2 channels. In particular, if ADPR hydrolysis facilitated channel closure, then disrupting catalysis would be expected to slow steady-state channel closing rate, or channel deactivation following sudden ligand removal, neither of which was observed here (*Figure 7E* and *Figures 4D*, *5D* and *6G*, respectively). Thus, the catalytic cycle of nvTRPM2 is entirely uncoupled from pore gating.

Of note, the steady-state mean burst durations (1–2 s) measured in the presence of saturating ligand (*Figure 7E*) were at least tenfold longer than the deactivation time constants upon ligand removal (*Figures 4D* and *6G*). Insofar as the latter is a measure of mean burst duration at zero ligand concentration, these findings suggest an even more pronounced [ADPR] dependence of burst durations for nvTRPM2 than reported for the human channel (*Tóth et al., 2014*), implying that ADPR bound to the activating site (likely the N-terminal site) remains readily exchangeable even in the open state.

If not required for channel gating, then what role did the robust ADPRase activity of ancient TRPM2 channels play? One possible explanation is that the channels used this strategy to rapidly clear away their activating ligand, thereby limiting Ca$^{2+}$ influx in time. Indeed, invertebrate TRPM2 channels do not possess an intrinsic mechanism for inactivation (*Figure 8B–C*). Moreover, Ca$^{2+}$ influx through their highly Ca$^{2+}$ permeable pores provides a positive feedback by supplying one of the two activating ligands (*Zhang et al., 2018*). On the other hand, prolonged activation of a channel as Ca$^{2+}$ permeable as the nvTRPM2 pore (*Zhang et al., 2018*) would be expected to result in Ca$^{2+}$ overload and cell death. Thus, degradation of the other essential ligand, ADPR, would seem a plausible strategy for self-regulation. By boosting TRPM2 ADPRase activity (*Figures 2C* and *3B*), Mg$^{2+}$ influx through the open pore might contribute important negative feedback to the regulation of invertebrate TRPM2 channel activity. Directly addressing this possibility in live cells of various invertebrate species is beyond the scope of the present study, but given that cytosolic ADPR levels are in

the low micromolar range (e.g. *Heiner et al., 2006*), the measured $k_{cat}$ for nvTRPM2 (~10 s$^{-1}$/subunit at pH = 7.1 and 2 mM Mg$^{2+}$;~40 s$^{-1}$/subunit at basic pH and high [Mg$^{2+}$] relevant to marine organisms), and even for srTRPM2 (~2 s$^{-1}$/subunit), are compatible with such a hypothesis. For example, in a *Salpingoeca rosetta* cell (cell volume ~10$^{-15}$ liter; *Dayel et al., 2011*), 10 μM ADPR corresponds to ~10$^4$ molecules, which could be cleared by srTRPM2 channels within 1–2 min, assuming only ~25 tetrameric channels/cell.

If so, then why did this enzymatic activity get lost through the course of evolution, and what other mechanism might have taken over its role? Based on sequence alignment, the mutations in the Nudix box sequence which abolished catalysis in TRPM2 channels (*Figure 1*, *right*), and the specific changes in the pore sequence that cause inactivation (*Figure 1*, *left*), appeared at about the same evolutionary time, between chordates and vertebrates. Indeed, functional experiments confirmed stable pores for the two invertebrate channels (*Figure 8B–C*), but inactivating pores for the two vertebrate channels (*Figure 8D–E*) tested in this study. It is therefore possible that pore inactivation of TRPM2 channels emerged to provide an alternative mechanism for turning off channel activity. An advantage of the latter mechanism might be that it allows the time window of Ca$^{2+}$ influx through TRPM2 channels to be regulated independently of the degradation time course of cytosolic ADPR, which might have acquired additional functions as a signaling molecule in vertebrates. Further studies in intact cells will be required to test this hypothesis.

# Materials and methods

## Key resources table

| Reagent type (species) or resource | Designation | Source or reference | Identifiers | Additional information |
|---|---|---|---|---|
| Gene (*Salpingoeca rosetta* TRPM2) | srTRPM2, srNUDT9-H | General Biosystems, Inc. | NCBI: XP_004993318 | species-optimized synthetic genes |
| Gene (*Nematostella vectensis* TRPM2) | nvTRPM2, nvNUDT9-H | General Biosystems, Inc. | NCBI: XP_001622235 | species-optimized synthetic genes |
| Strain, strain background (*Escherichia coli* BL21 (DE3)) | *E.coli* BL21 (DE3) | New England BioLabs | C2527H | |
| Cell line (*Spodoptera frugiperda*) | Sf9 | ATCC | CRL-1711 | authenticated and mycoplasma free by vendor |
| Cell line (*Homo sapiens*) | HEK293S GnTI- | ATCC | CRL-3022 | authenticated and mycoplasma free by vendor |
| Biological sample (*Xenopus laevis*) | *Xenopus laevis* oocytes | African Reptile Park | RRID: NXR_0.0080 | mandyvorster@xsinet.co.za |
| Commercial assay or kit | HiSpeed Plasmid Midi Kit | Qiagen | 12643 | |
| Commercial assay or kit | QuickChange II Mutagenesis Kit | Agilent Technologies | 200524–5 | |
| Commercial assay or kit | mMESSAGE mMACHINE T7 Transcription Kit | ThermoFisher | AM1344 | |
| Commercial assay or kit | NHS Activated Sepharose 4 Fast Flow | GE Healthcare | 17-0906-01 | |
| Commercial assay or kit | Superose 6 Increase 10/300 GL | GE Healthcare | 29-0915-96 | |

*Continued on next page*

*Continued*

| Reagent type (species) or resource | Designation | Source or reference | Identifiers | Additional information |
|---|---|---|---|---|
| Commercial assay or kit | Strep-Tactin MacroPrep cartridge | IBA GmbH | 2-1538-001 | |
| Chemical compound, drug | Avidin | IBA GmbH | 2-0204-015 | |
| Chemical compound, drug | D-Desthiobiotin | IBA GmbH | 2-1000-002 | |
| Chemical compound, drug | Adenosine 5'-diphosphoribose sodium (ADPR) | Sigma-Aldrich | A0752 | |
| Chemical compound, drug | 2,2-didecylpropane-1,3-bis-b-D-malto pyranoside (LMNG) | Anatrace | NG310 | |
| Chemical compound, drug | Cholesteryl hemisuccinate (CHS) | Anatrace | CH210 | |
| Chemical compound, drug | Digitonin | Sigma-Aldrich | D141 | |
| Chemical compound, drug | DMEM:F12, 1:1 Mixture with 3.151 g/L glucose, HEPES, and L-glutamine | LONZA | 12–719F | |
| Chemical compound, drug | Fetal Bovine Serum (South America Origin), EU approved | LONZA | ECS0180L | Heat inactivated at 56°C for 30 min. |
| Chemical compound, drug | sf-900 II SFM medium | Gibco | 10902088 | |
| Chemical compound, drug | Freestyle 293 medium | Gibco | 12338018 | |
| Chemical compound, drug | Cellfectin II | Invitrogen | 10362100 | |
| Chemical compound, drug | Antibiotic-Antimycotic (100X) | Gibco | 15240062 | |
| Chemical compound, drug | Isopropyl β- D —1-thiogala ctopyranoside (IPTG) | Invitrogen | 15529019 | |
| Chemical compound, drug | Bluo-GAL | Invitrogen | 15519028 | |
| Chemical compound, drug | Tetracycline | Sigma-Aldrich | T7660 | |
| Chemical compound, drug | Kanamycin | Sigma-Aldrich | K1377 | |

*Continued on next page*

*Continued*

| Reagent type (species) or resource | Designation | Source or reference | Identifiers | Additional information |
|---|---|---|---|---|
| Chemical compound, drug | Chloramphenicol | Sigma-Aldrich | C0378 | |
| Chemical compound, drug | Gentamicin | Sigma-Aldrich | G1397 | |
| Chemical compound, drug | Collagenase type II | Gibco | 17107–0125 | |
| Chemical compound, drug | PreScission Protease | GE Healthcare | 27084301 | |
| Software, algorithm | Pclamp9 | Molecular Devices | RRID: SCR_011323 | |
| Software, algorithm | Pymol | PyMOL | http://www.pymol.org | |

## Molecular biology

For expression in frog oocytes, the full-length nvTRPM2 and srTRPM2 genes (XP_001622235 and XP_004993318) were sequence-optimized for *Xenopus laevis* [RRID: NXR_0.0080], synthesized and incorporated into the pGEMHE vector (General Biosystems). The drTRPM2 gene in the pGEMHE vector was kindly provided by Dr. Seok-Yong Lee (Duke University). The hsTRPM2-pGEMHE construct was constructed as described (*Csanády and Törocsik, 2009*). cDNA was transcribed from linearized (NheI) pGEMHE-TRPM2 using T7 polymerase, and cRNA stored at −80°C. The DNA constructs used for mammalian expression of full-length nvTRPM2 were described previously (*Zhang et al., 2018*); the DNA constructs for mammalian expression of full-length srTRPM2 were prepared identically. For bacterial expression of wild-type nvNUDT9-H and srNUDT9-H, the DNA sequences encoding nvTRPM2 residues 1271–1551 and srTRPM2 residues 1215–1494 were sequence-optimized for *E.coli*, synthesized with added C-terminal Twin-Strep-tags (General Biosystems), and incorporated into the pJ411 vector (DNA2.0). The double mutations E1443I/F1444L and E1446K/E1447K were introduced into pGEMHE-nvTRPM2 and pJ411-nvNUDT9-H using the Stratagene QuickChange II Site-Directed Mutagenesis Kit (Agilent Technologies). All mutants were sequence-verified (LGC Genomics GmbH).

## Protein expression and purification

Full-length nvTRPM2 and srTRPM2 proteins were expressed in HEK 293S GnTI⁻ cells (ATCC CRL-3022, authenticated and mycoplasma free by vendor) and purified as described in detail previously for nvTRPM2 (*Zhang et al., 2018*). In brief, high titer recombinant baculoviruses generated in Sf9 insect cells (ATCC CRL-1711, authenticated and mycoplasma free by vendor), carrying the nv/srTRPM2 gene with a C-terminal GFP tag, were added to HEK 293S GnTI⁻ cells. After a 12 hr incubation at 37°C, protein expression was induced by 10 mM sodium butyrate for 48 hr at 30°C. Cells were harvested by centrifugation, resuspended, homogenized, and solubilized with 1% 2,2-didecyl-propane-1,3-bis-β-D-maltopyranoside (LMNG) and 0.1% Cholesteryl hemisuccinate (CHS). After centrifugation at 50,000 g for 1 hr, the supernatant was bound to GFP nanobody-coupled resin, and washed extensively to exchange LMNG and CHS with 0.06% digitonin. The nv/srTRPM2 protein was released from the resin using PreScission protease, and further purified by gel filtration on a Superose 6 10/300 column (GE Healthcare).

Isolated srNUDT9-H and (WT and mutant) nvNUDT9-H domains were expressed in *E.coli* BL21 (DE3) and purified using a protocol similar to that described previously for human NUDT9-H (*Iordanov et al., 2016*). In brief, bacterial cultures were grown at 25°C in Luria-Bertani medium supplemented with 50 µg/ml kanamycin and induced with 0.1 mM isopropyl-β-D-1-thio-galactopyranoside (IPTG) upon reaching $OD_{600}$ of ~0.5. After overnight incubation at 25°C, the cells were

harvested and lysed by sonication in 100 mM Tris (pH 8.5 with HCl)/150 mM NaCl, supplemented with Halt Protease Inhibitor Cocktail (Thermo Scientific). The cleared supernatant was subjected to Strep-Tactin affinity chromatography following the manufacturer's instructions (IBA GmbH). The affinity-purified proteins were then concentrated (10,000 MWCO Vivaspin, Sigma-Aldrich) and passed through a gel filtration column (Superdex 200 10/300 GL, GE Healthcare), and the main peak fractions were isolated. Affinity tags were not removed. Protein purity was visually checked by SDS PAGE, and protein identity confirmed by the band position and the main peak position on the gel filtration profile (*Figure 2—figure supplement 1*, *Figure 3—figure supplement 1*). Protein concentrations were determined spectrophotometrically using theoretical molar extinction coefficients ($\varepsilon_0$ = 58,320 $M^{-1}cm^{-1}$ for nvNUDT9-H and 66,100 $M^{-1}cm^{-1}$ for srNUDT9-H at 280 nm); the yield was ~5 mg/L of culture for srNUDT9-H, and also for both WT and mutant nvNUDT9-H, proteins. The purified proteins were flash-frozen in liquid nitrogen and stored at −20°C until used.

## Enzymatic activity assay

The ADPRase activities of purified srTRPM2, nvTRPM2, srNUDT9-H, and WT and mutant nvNUDT9-H were assessed through colorimetric detection of inorganic phosphate ($P_i$) liberated from the ADPR cleavage products AMP and ribose-5-phosphate (R5P) by co-applied alkaline phosphatase (AP) (*Rafty et al., 2002*). Because AP liberates $P_i$ from AMP and R5P, but not from intact ADPR, 2 moles of $P_i$ are released per mole of ADPR hydrolyzed. For nv/srTRPM2, 15 µl volumes of reaction buffer (20 mM Tris (pH 8.0), 150 mM NaCl, 10 mM $MgCl_2$, 0.06% digitonin) containing 0.4 nM purified protein, 2–3 U bovine AP, and 10–300 µM ADPR were incubated for 10 min at room temperature. For NUDT9-H proteins, 150 µl volumes of reaction buffer (50 mM MES, HEPES, or Tris (pH 5.7 to 8.5), 0-320 mM $MgCl_2$) containing 0.5, 10, 50, or 500 nM purified protein, 2–3 U bovine AP, and 5–320 µM ADPR were incubated for 4–5 min at room temperature. The reactions were stopped and the liberated $P_i$ visualized by adding 85 (nv/srTRPM2) or 850 (NUDT9-H proteins) µl coloring solution (6:1 vol/vol ratio mixture of 0.42% ammonium molybdate tetrahydrate in 1N $H_2SO_4$ and 10% L-ascorbic acid) followed by incubation for 20 min at 45°C. Absorption was measured at 820 nm (NanoPhotometer P300, Implen GmbH) and compared to that of a standard curve. The coloring solution was freshly made, and the standard curve (1–2000 µM $KH_2PO_4$) obtained, daily. Reactions with 100 µM AMP (instead of ADPR) served as positive controls. All reagents were from Sigma-Aldrich.

The molecular turnover rates ($k_{cat}$) for full-length nv/srTRPM2 were calculated from the $P_i$ liberated from 2 mM ADPR in 10 min by 10–140 nM protein (subunit). $k_{cat}$ values for NUDT9-H proteins were calculated after 5–10 min reactions using 20 (WT sr), 0.5 (WT nv), 50 (E1443I/F1444L nv) or 500 (E1446K/E1447K nv) nM protein and saturating ADPR. Not more than 10% of the initial substrate was hydrolyzed during the reactions, that is saturation was maintained throughout the incubation time. To test hydrolytic activity of WT nvNUDT9-H in the absence of $Mg^{2+}$, $MgCl_2$ was omitted from the reaction buffer with or without addition of 100 µM 1,2-cyclohexylenedinitrilotetraacetic acid (CDTA). The ability of WT nvNUDT9-H to hydrolyze AMPCPR was tested by substituting ADPR in the reaction mixture with 100 µM AMPCPR. All assays were performed at least in triplicates, and data are displayed as mean ± SEM.

## Visualization of enzymatic activity by thin-layer chromatography (TLC)

20 µL reaction mixtures containing 10 mM ADPR and 0.1 µM wild-type nvNUDT9-H, 3 µM nvNUDT9-H E1443I/F1444L, 40 µM nvNUDT9-H E1446K/E1447K, or 1 µM wild-type srNUDT9-H were incubated for 1 hr at room temperature in 50 mM Tris (pH 8.5) supplemented with 16 mM $MgCl_2$. 1 µl aliquots of the reaction mixtures were placed on Polygram SIL G/UV254 plates (Macherey-Nagel, Düren, Germany), dried and developed in 0.2 M $NH_4HCO_3$ in ethanol:water 7:3 (vol/vol). 10 mM ADPR and 10 mM AMP enzyme-free controls were treated identically and used to visualize the nucleotides' positions on the TLC sheet, and to monitor the spontaneous degradation of ADPR. Nucleotides were visualized under UV light.

## Functional expression of TRPM2 orthologs in Xenopus laevis oocytes

*Xenopus laevis* oocytes were isolated, collagenase digested, injected with 0.1–10 ng of wild-type or mutant srTRPM2, nvTRPM2, drTRPM2, or hsTRPM2 cRNA, and stored at 18°C. Recordings were done 1–3 days after injection.

## Electrophysiology

Excised inside-out patch-clamp recording of TRPM2 currents was done as described (*Zhang et al., 2018*). Pipette solution contained (in mM) 140 Na-gluconate, 2 Mg-gluconate$_2$, 10 HEPES, 1 EGTA (pH = 7.4 with NaOH). The pipette electrode was placed into a 140 mM NaCl based solution carefully layered on top (*Csanády and Törocsik, 2009*). Bath solution contained (in mM) 140 Na-gluconate, 2 mM Mg-gluconate$_2$, 10 mM HEPES (pH 7.1 with NaOH), and either 1 mM EGTA (to obtain 'zero' (~1 nM) Ca$^{2+}$), or 1 mM Ca-gluconate$_2$ (to obtain 125 µM free [Ca$^{2+}$]). In such symmetrical Na-gluconate-based solutions cytosolic exposure to ADPR+Ca$^{2+}$ elicits large Na$^+$ currents in patches excised from oocytes pre-injected with nvTRPM2 cRNA (*Zhang et al., 2018*), whereas no currents are seen in patches excised from non-injected or water-injected oocytes (*Figure 4—figure supplement 3*; see also *Csanády and Törocsik, 2009*). For segments of recording in zero cytosolic Mg$^{2+}$ (*Figures 4B* and *7A*), Mg-gluconate$_2$ was omitted from the bath solution and 0.1 mM CDTA was added; for such recordings, Mg-gluconate$_2$ and EGTA were omitted from the pipette solution. Recordings were obtained at 25°C at a membrane potential of −20 mV, under continuous superfusion of the cytosolic patch surface. Solution exchange time constant was <50 ms. Currents were digitized at 10 kHz, filtered at 2 kHz and recorded to disk. Na$_2$-ADPR was obtained from Sigma, Na-AMPCPR was synthesized by Dr. Krzysztof Felczak (University of Minnesota) as described (*Pankiewicz et al., 1997*).

## Kinetic analysis of macroscopic current recordings

Macroscopic current relaxations were least-squares fitted by decaying single exponential functions. Fractional currents (*Figure 6F*) were calculated by dividing mean current in a test segment by mean current in 100 µM ADPR in the same patch.

## Kinetic analysis of microscopic current recordings

For steady-state single-channel kinetic analysis, recordings with well resolved unitary transitions, from patches containing 1–12 active channels, were digitally filtered at 200 Hz, baseline subtracted, and idealized by half-amplitude threshold crossing. To extract mean burst ($\tau_b$) and interburst duration ($\tau_{ib}$), the set of dwell-time histograms obtained for all conductance levels was fitted by a C$_{slow}$-O-C$_{fast}$ model (*Figure 7—figure supplement 1*) using maximum likelihood (*Csanády, 2000*), and $\tau_b$ and $\tau_{ib}$ calculated from the fitted rate constants as described (*Tóth et al., 2014*). Of note, the choice of a C-O-C model is not unique for describing the observed gating pattern, as a C-C-O model is identically suitable. However, whereas the extracted transition rates obviously depend on the chosen model, the calculated mean burst and interburst durations are model-independent.

The number of active channels in the patch (N) was estimated as the maximum number of simultaneously open channels (N'). The likelihood of the presence of additional active channels in the patch (i.e. N > N') was evaluated using a statistical test based on a comparison of channel opening rate (1/$\tau_{ib}$) with the cumulative time spent at level N' (*Csanády, 2000*). For all patches included in the analysis, the possibility of N > N' could be excluded with high confidence (p < 0.001).

## Estimation of unitary current amplitudes in microscopic recordings

All-points histograms (*Figure 4—figure supplement 1*, *Figure 4—figure supplement 2*) were fitted by sums of Gaussian functions and unitary current amplitudes calculated as the mean distance between adjacent peaks.

## Acknowledgements

We thank Ying Yin and Seok-Yong Lee for the pGEMHE-drTRPM2 plasmid. Supported by an International Early Career Scientist grant from the Howard Hughes Medical Institute and Lendület grant LP2017-14/2017 from the Hungarian Academy of Sciences to LC, and a New National Excellence Program (ÚNKP) award of the Ministry of Human Capacities of Hungary to Semmelweis University. BT is supported by postdoctoral ÚNKP grants 17–4-I-SE-61 and 18–4-SE-132. BT and AS are János Bolyai Research Fellows.

# Additional information

## Competing interests
László Csanády: Reviewing editor, *eLife*. The other authors declare that no competing interests exist.

## Funding

| Funder | Grant reference number | Author |
| --- | --- | --- |
| Howard Hughes Medical Institute | International Early Career Scientist Award | László Csanády |
| Magyar Tudományos Akadémia | LP2017-14/2017 | László Csanády |
| Ministry of Human Capacities of Hungary | ÚNKP 17-4-I-SE-61 | Balázs Tóth |
| Magyar Tudományos Akadémia | Bolyai Research Fellowship | Balázs Tóth |
| Ministry of Human Capacities of Hungary | ÚNKP-FIKP | László Csanády |
| Ministry of Human Capacities of Hungary | ÚNKP 18-4-SE-132 | Balázs Tóth |

The funders had no role in study design, data collection and interpretation, or the decision to submit the work for publication.

## Author contributions
Iordan Iordanov, Andras Szollosi, Data curation, Formal analysis, Validation, Investigation, Visualization, Methodology, Writing—original draft; Balázs Tóth, Data curation, Formal analysis, Validation, Investigation, Visualization, Methodology; László Csanády, Conceptualization, Data curation, Software, Supervision, Funding acquisition, Validation, Visualization, Methodology, Writing—original draft, Project administration, Writing—review and editing

## Author ORCIDs
Iordan Iordanov (iD) http://orcid.org/0000-0001-8251-5857
Balázs Tóth (iD) http://orcid.org/0000-0002-1257-2597
Andras Szollosi (iD) http://orcid.org/0000-0002-5570-4609
László Csanády (iD) http://orcid.org/0000-0002-6547-5889

## Ethics
Animal experimentation: This study was performed in strict accordance with the recommendations in the Guide for the Care and Use of Laboratory Animals of the National Institutes of Health. All of the animals were handled according to approved institutional animal care and use committee (IACUC) protocols of Semmelweis University (last approved 06-30-2016, expiration 06-30-2021).

## Decision letter and Author response
Decision letter https://doi.org/10.7554/eLife.44556.028
Author response https://doi.org/10.7554/eLife.44556.029

# Additional files

## Supplementary files
• Transparent reporting form
DOI: https://doi.org/10.7554/eLife.44556.026

## Data availability

All data generated or analyzed during this study are included in the manuscript figures and supplementary figures. All methods are described in detail in Materials and Methods. Source data files are attached for all figures.

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
