## [Decision Letter]

Thank you for submitting your article "Enzyme activity and selectivity filter stability of ancient TRPM2 channels were simultaneously lost in early vertebrates" for consideration by *eLife*. Your article has been reviewed by three peer reviewers, including Leon D Islas as the Reviewing Editor and Reviewer #1, and the evaluation has been overseen by Richard Aldrich as the Senior Editor. The following individuals involved in review of your submission have agreed to reveal their identity: Tibor Rohacs (Reviewer #2); David E Clapham (Reviewer #3).

The reviewers have discussed the reviews with one another and the Reviewing Editor has drafted this decision to help you prepare a revised submission.

Summary:

TRPM2 channels belong to a special category of ion channels coupled to enzymes name chanzymes. Your manuscript addresses important evolutionary aspects of the function of TRPM2 channels. The key finding is that the Nudix homology domain of invertebrates is catalytically active, which is consistent with their Nudix box sequence containing two key glutamate residues important for catalytical activity of this enzyme family. Vertebrate TRPM2 channels lack these residues and they have been shown earlier to be catalytically inactive. In this study, you used a combination of biochemistry and electrophysiology to investigate the biophysical and enzymatic properties of TRPM2 from two invertebrate organisms. You provide convincing evidence that the multiple pore/NUDT9-H mutations of invertebrate TRPM2 lead to proteins with vastly different function. Invertebrate TRPM2, particularly nvTRPM2, is clearly a functional ADPRase.

Essential revisions:

1) In the subsection “srTRPM2 is also an active enzyme”, the Mg^2+^ dependence of ADPR hydrolysis of the isolated srNUDT9 domain is given as 1 mM, however this cannot be estimated from the data since there are only two Mg^2+^ concentration points.

2) Please also indicate if the full srTRPM2 channel was tested for catalytic activity.

3) Since the time constant of solution change in inside out patches is 50 ms and the time constants of current decay upon ADPR removal are barely two times that, I do not think that meaningful conclusions can be obtained from comparison of the decay time constants, especially when comparing to the burst lengths.

4) Regarding the calculation of the burst-length and inter-burst intervals, the authors calculate this parameters from open and closed dwell times obtained from multi-channel patches. Since in this conditions the individual open and closed times are meaningless, the authors correctly only calculate burst lengths. However, the inter-burst intervals are also very difficult to interpret and the fitting to a 3-state model is not unique. Since the burst length calculation in multichannel patches is highly dependent on a correct estimation of the number of channels, the authors should clearly explain how this estimate was obtained and what are the limitations of the analysis as it was performed.

5) The excised patch data in some of the figures are not quantified. I recommend that in Figures 4 and 5 current amplitudes are quantified, and the number of experiments is unambiguously stated. In its current form, it is not clear if the n for the time constants in Figure 4C and in Figure 5C refer to number of patches tested, or number of individual measurements i.e. for Figure 4, does the tau for panel 4B counts as n=6 for Mg^2+^ or n=1? Also in Figure 8, even though the difference between invertebrate and drTRPM2 is striking based on the representative traces, some quantification is necessary.

6) Can the authors rescue catalytic activity of hsTRPM2 with changing residues in the nudix domain to those in invertebrates? In other words is the difference in the nudix box the sole reason for the lack of activity or did vertebrate channels accumulate other mutations also that destroy catalytic activity?

7) It is not clear that srNUDT9-H is an enzyme given the low *k*_cat_. This protein binds ADPR with a micromolar affinity, however the rate of catalysis is low and little affected by pH or Mg^2+^. Knowing that background ADPR hydrolysis occurs in their experimental setting (Figure 4A) and that non-functional enzymes have *k*_cat_ values around 0.4 (Figure 6B), how confident are you that srNUDT9-H is a bona fide enzyme (*k*_cat_ values around 1.5-2.0) compared to nvNUDT9-H (*k*_cat_ near 40)? The authors might want to defend their conclusion that this is an enzyme or discuss the possibility that it is non-functional.

---

## [Author Response]

Essential revisions:1) In the subsection “srTRPM2 is also an active enzyme”, the Mg^2+^ dependence of ADPR hydrolysis of the isolated srNUDT9 domain is given as 1 mM, however this cannot be estimated from the data since there are only two Mg^2+^ concentration points.

We initially focused our attention primarily onto nvTRPM2 since we had previously characterized its structure and function in great detail (Zhang et al., 2018). We therefore contented ourselves to simply showing that srTRPM2 is also catalytically active. The reviewers were quite right that the K_1/2_ for Mg^2+^ binding cannot be reliably determined from two data points: our rough estimate was based on the assumption of a simple binding curve which in retrospect turned out to be incorrect. We have now performed complete titrations of srNUDT9-H with Mg^2+^ at two different fixed pH values (7.1 and 8.5). The curves, shown in Figure 3B, reveal a more complex dependence of *k*_cat_ on [Mg^2+^], suggesting the presence of high- and low-affinity binding sites. The mechanistic interpretation of these findings is presented in the first paragraph of the subsection “srTRPM2 is also an active enzyme”, and its possible biological relevance in the fourth paragraph of the Discussion.

2) Please also indicate if the full srTRPM2 channel was tested for catalytic activity.

We did not initially test enzymatic activity for full-length srTRPM2, because this protein preparation is unstable, showing clear signs of precipitation during size-exclusion chromatography. Nevertheless, we have now attempted some enzymatic assays on the full-length protein. These assays clearly confirmed that full-length srTRPM2 is also active, and its estimated *k*_cat_ value (which, considering the above limitations, has to be taken as a lower estimate) is consistent with the *k*_cat_ observed for the isolated srNUDT9-H domain. These results are now shown in Figure 3C.

3) Since the time constant of solution change in inside out patches is 50 ms and the time constants of current decay upon ADPR removal are barely two times that, I do not think that meaningful conclusions can be obtained from comparison of the decay time constants, especially when comparing to the burst lengths.

The reviewers correctly note that the macroscopic relaxation time constants might be slightly overestimating the true time constants of current decay which would be observed with an extremely fast solution exchange system. However, even given these limitations both of our conclusions regarding these time constants are solid. (i) Mg^2+^ removal does certainly not prolong these time constants, as would be expected if ADPR hydrolysis accelerated channel closure. We now use this more careful wording in the second paragraph of the subsection “Disrupting catalysis at the NUDT9-H domain does not affect macroscopic gating parameters of nvTRPM2W”. (ii) Our comparison between these time constants and steady-state burst lengths is similarly safe: we conclude (Discussion, seventh paragraph) that steady-state bursts are at least tenfold longer than the relaxation time constants. If the measured relaxation time constants overestimate the true values, then this conclusion holds even stronger.

4) Regarding the calculation of the burst-length and inter-burst intervals, the authors calculate this parameters from open and closed dwell times obtained from multi-channel patches. Since in this conditions the individual open and closed times are meaningless, the authors correctly only calculate burst lengths. However, the inter-burst intervals are also very difficult to interpret and the fitting to a 3-state model is not unique. Since the burst length calculation in multichannel patches is highly dependent on a correct estimation of the number of channels, the authors should clearly explain how this estimate was obtained and what are the limitations of the analysis as it was performed.

The algorithm used here for burst analysis is extremely robust and, since its original publication ~20 years ago, has been successfully used by several groups (working mainly in the CFTR field), resulting in >50 published papers. All of the concerns raised by the reviewers were carefully investigated and discussed in the original study (Csanády, 2000) and in following publication (Csanády et al., 2000). We have therefore not provided a detailed description of the algorithm here. On the other hand, we do realize that multi-channel kinetic analysis is a specialized technique which will be unfamiliar to many readers, so we have added some clarifying explanations, without going into very great depth.

The reviewers' concerns touch on three independent issues:

i) Interpretation of dwell times in multi-channel patches:

“…the authors calculate this parameters from open and closed dwell times obtained from multi-channel patches… in this conditions the individual open and closed times are meaningless…”

"Open" and "closed" dwell times are indeed meaningless under such conditions. Instead dwell times can be calculated for each conductance level, i.e., dwell times during which 0, 1, 2,.… N channels are open (as shown in Figure 7—figure supplement 1). Construction of such dwell-time histograms for each conductance level simply requires that the individual gating transitions remain well resolved, which was certainly the case here, as shown in Figure 7A-C (insets). How can such dwell-time histograms be interpreted in terms of a single-channel gating scheme? With the assumption of N identical and independent channels the entire system of N channels can be considered as a single macro-system which can exist in a finite number of macro-states, the transition rates among which are well defined functions of the single-channel transition rate constants (Horn and Lange, 1983). Therefore, fitting the set of N+1 dwell-time histograms (for conductance levels 0, 1,… N) of a multi-channel patch with the underlying single channel gating scheme is just as meaningful as fitting the closed- and opentime histograms of a single-channel patch. Indeed, because the time constants and fractional amplitudes of the exponential components of each dwell-time histogram, as well as their relative scaling, are well-defined functions of the same set of single-channel transition rates, such a data set provides robust constraints for determining the latter.

ii) Identifiability of the underlying gating model from multi-channel patches:

“… the inter-burst intervals are also very difficult to interpret… and the fitting to a 3-state model is not unique.”

Dwell-times at the highest conductance level (i.e., with all channels open) were single-exponentially distributed, whereas at all other conductance levels the dwell-time distributions showed two exponential components (Figure 7—figure supplement 1). Such a pattern of dwell-time histograms uniquely identifies a bursting gating pattern with one open (O) and two closed (C_fast_, C_slow_) single-channel states. At conductance level N, the macrosystem of N such channels can exist in only one macro-state (all channels are in state O), giving rise to a single exponential distribution (the time constant of which equals the single-channel open time divided by N). Conversely, at conductance level N-1 the macro-system can exist in two possible macro-states: either N-1 channels are in state O and 1 channel in state C_slow_ (giving rise to the slower component of the distribution), or N-1 channels are in state O and 1 channel in state C_fast_ (giving rise to the faster component of the distribution). Finally, at levels below N-1 there are increasing numbers of possible macro-states, but those in which more than 1 channel simultaneously dwells in C_fast_ will be negligibly visited. Therefore, at any conductance level k (<N) only two components will be detected in practice: a slow component corresponding to the macro-state with k channels in state O and N-k channels in state C_slow_, and a fast component corresponding to the macro-state with k channels in state O, N-k-1 channels in state C_slow_, and 1 channel in state C_fast_. This pattern is very obvious in the histograms shown in Figure 7—figure supplement 1. In summary, the presence of one exponential component at level N and two exponential components at level N-1 clearly identifies one open and two closed states, just as unambiguously as a single-exponential open-time distribution and a double-exponential closed-time distribution would do in a single-channel patch. A brief summary of this explanation is now included in the subsection “Catalytic cycle at the NUDT9-H domain is not coupled to gating conformational changes”.

The choice of a C-O-C model is indeed not unique for describing the observed gating pattern: a C-C-O model would be identically suitable. However, whereas the extracted transition rates obviously depend on the chosen model, the calculated mean burst and interburst durations are model-independent. This is now stated in Materials and methods (subsection “Kinetic analysis of microscopic current recordings”).

iii) Estimation of the correct number of channels in a multi-channel patch:

“Since the burst length calculation in multichannel patches is highly dependent on a correct estimation of the number of channels, the authors should clearly explain how this estimate was obtained and what are the limitations of the analysis as it was performed.”

Correct estimation of the number of active channels (N) is just as important for recordings with superimposed channel openings, as it is for the case of a patch in which no superimposed channel openings are seen. The approaches used to estimate N are also similar in both cases. In general, N can be estimated with high confidence when the P_o_ is high, but not when the P_o_ is low. For the nvTRPM2 constructs studied here P_o_ was high (~0.8), and therefore, estimating N was not a problem.

The number of active channels in the patch (N) was estimated as the maximum number of simultaneously open channels (N'). The likelihood of the presence of additional active channels in the patch (i.e., the hypothesis of N>N') was evaluated using a statistical test based on a comparison of channel opening rate (1/τ_ib_) with the cumulative time spent at level N' (Csanády et al., 2000). For all patches included in the analysis the possibility of N>N' could be excluded with high confidence (p<0.001). This is now stated in Materials and methods (subsection “subsection “Kinetic analysis of microscopic current recordings”).

5) The excised patch data in some of the figures are not quantified. I recommend that in Figures 4 and 5 current amplitudes are quantified.

Done. See new Figure 4C and 5C.

And the number of experiments is unambiguously stated. In its current form, it is not clear if the n for the time constants in Figure 4C and in Figure 5C refer to number of patches tested, or number of individual measurements i.e. for Figure 4, does the tau for panel 4B counts as n=6 for Mg^2+^ or n=1?

The n values represent the number of patches, as is now clearly stated in each figure legend. (I.e., the average tau value obtained under a given condition from the trace in Figure 4B counts as n=1.)

Also in Figure 8, even though the difference between invertebrate and drTRPM2 is striking based on the representative traces, some quantification is necessary.

We now report mean ± SEM tau values for drTRPM2 and hsTRPM2 in the text (subsection “Invertebrate TRPM2 pores are stable whereas vertebrate TRPM2 channels inactivate”, last paragraph).

6) Can the authors rescue catalytic activity of hsTRPM2 with changing residues in the nudix domain to those in invertebrates? In other words is the difference in the nudix box the sole reason for the lack of activity or did vertebrate channels accumulate other mutations also that destroy catalytic activity?

We could not ascertain whether the mutations in the key Nudix box positions (Figure 1, right, asterisks) are solely responsible for the loss of catalysis in vertebrate TRPM2 channels, as we were unable to obtain hsNUDT9-H constructs with "revertant" Nudix box motifs (Nudix box: REFRQE or REFGEE) in soluble form. Thus, although the Nudix box mutations are clearly *sufficient* to abrogate enzymatic activity (Figure 6B, Perraud et al., 2003), it remains possible that in vertebrate TRPM2 channels additional mutations have accumulated which are also incompatible with catalysis (cf., Kuhn et al., 2015). This is now discussed in the first paragraph of the Discussion.

7) It is not clear that srNUDT9-H is an enzyme given the low k_cat_. This protein binds ADPR with a micromolar affinity, however the rate of catalysis is low and little affected by pH or Mg^2+^. Knowing that background ADPR hydrolysis occurs in their experimental setting (Figure 4A) and that non-functional enzymes have k_cat_ values around 0.4 (Figure 6B), how confident are you that srNUDT9-H is a bona fide enzyme (k_cat_ values around 1.5-2.0) compared to nvNUDT9-H (k_cat_ near 40)? The authors might want to defend their conclusion that this is an enzyme or discuss the possibility that it is non-functional.

srNUDT9-H is clearly a functional enzyme. First, we now show that catalysis by srNUDT9-H is absolutely dependent on Mg^2+^, and also sensitive to pH (Figure 3B). Second, clearly separating even slow enzymatic activity from spontaneous background hydrolysis is not an issue, since those two processes can be discerned by increasing the amount of enzyme used in the assay. Indeed, even a *k*_cat_ as low as 0.04 s^-1^ (for the nvKK mutant, Figure 6B) can be reliably determined by our assay. We now show in Figure 6—figure supplement 1 the actual inorganic phosphate signal for all of the low-activity proteins, and compare them to the background signal. Third, we now show that full-length srTRPM2 is also enzymatically active (Figure 3C). Fourth, the *k*_cat_ of ~2 s^-1^ for srTRPM2 is taken per subunit, the *k*_cat_ for a tetrameric channel is ~8 s^-1^ (and might be further boosted by ~2-fold in very high Mg^2+^ (cf., Figure 3B, and Discussion, fourth paragraph).